# Topotactically transformable antiphase boundaries with enhanced ionic conductivity

Kun Xu[1,2,10] ✉, Shih-Wei Hung [3,4,10], Wenlong Si[1,5], Yongshun Wu[6], Chuanrui Huo[7], Pu Yu [6], Xiaoyan Zhong [3,4,8,9] ✉ & Jing Zhu[1,5] ✉

Engineering lattice defects have emerged as a promising approach to effectively modulate the functionality of devices. Particularly, antiphase boundaries (APBs) as planar defects have been considered major obstacles to optimizing the ionic conductivity of mixed ionic-electronic conductors (MIECs) in solid oxide fuel applications. Here our study identifies topotactically transformable APBs (tt-APBs) at the atomic level and demonstrates that they exhibit higher ionic conductivity at elevated temperatures as compared to perfect domains. In-situ observation at the atomic scale tracks dynamic oxygen migration across these tt-APBs, where the abundant interstitial sites between tetrahedrons facilitate the ionic migration. Furthermore, annealing in an oxidized atmosphere can lead to the formation of interstitial oxygen at these APBs. These pieces of evidence clearly clarify that the tt-APBs can contribute to oxygen conductivity as anion diffusion channels, while the topotactically non-transformable APBs cannot. The topotactic transformability opens the way of defect engineering strategies for improving ionic transportation in MIECs.

Control of defect formation in crystal materials is crucial for optimizing high-efficiency ceramic materials[1,2] and related functional devices[3,4]. The manipulation of crystal defects, such as antiphase boundaries[5,6], dislocation[7,8] and point defects[9,10] also proved to be effective in tuning or inducing diverse functionalities, e.g., conductivity[7], ferroelectricity[11], ferromagnetism[12,13], and piezoelectricity[9] in the past years. Mixed ionic-electronic conductors (MIECs), such as strontium ferrite oxides, are particularly attractive due to their significant oxygen permeability, structure stability, and chemical flexibility at high temperatures, as well as their reported mixed ionic and electronic conductivity[14–16]. Local defect structures[17,18] in MIECs have been widely investigated and demonstrated as an obstacle in oxygen ionic conductivity due to the stacking mismatch.

Among the various types of extended defects, the antiphase boundaries are commonly observed in ferrite films[19]. This boundary is defined as the interface between adjacent regions, where these regions exhibit a few sub-unit-cell shifts relative to each other along the out-of-plane direction in the adjacent domains. These phase boundaries disrupt the continuity of the layered structure in the in-plane direction and exert a significant influence on the material properties[19–21].

To enhance the functionalities of oxide materials, topotactic transformations have been used, involving delicate structural changes between related crystal structures induced by a loss, replacement, or gain of atoms while inheriting crystallographic similarities[22], for the performance optimization of devices[23]. Triggered under external excitation such as electric biasing, thermal heating, and atmosphere

[1]National Center for Electron Microscopy in Beijing, School of Materials Science and Engineering, Tsinghua University, Beijing 100084, PR China. [2]Department of Mechanical Engineering, Stanford University, Palo Alto 94305, USA. [3]TRACE EM Unit and Department of Materials Science and Engineering, City University of Hong Kong, Kowloon 999077 Hong Kong SAR, PR China. [4]City University of Hong Kong Matter Science Research Institute (Futian, Shenzhen), Shenzhen 518048, PR China. [5]Ji Hua Laboratory, Foshang, Guangdong 0757, PR China. [6]State Key Laboratory of Low Dimensional Quantum Physics and Department of Physics, Tsinghua University, Beijing 100084, PR China. [7]Beijing Advanced Innovation Center for Materials Genome Engineering, Department of Physical Chemistry, University of Science and Technology Beijing, Beijing 100083, PR China. [8]Nanomanufacturing Laboratory (NML), Shenzhen Research Institute, City University of Hong Kong, Shenzhen 518057, PR China. [9]Chengdu Research Institute, City University of Hong Kong, Chengdu 610200, PR China. [10]These authors contributed equally: Kun Xu, Shih-Wei Hung. ✉e-mail: kunxuem@stanford.edu; xzhong25@cityu.edu.hk; jzhu@mail.tsinghua.edu.cn

environment, topotactic transformations with distinct oxygen variation in metal oxides, can lead to rich physical properties, such as metal-to-insulator transition, magnetic and transport variation, etc[24]. In particular, topotactic transformation in MIECs has been recognized as a way to modulate the oxygen migration channels, where the collective ionic propagation can be activated to improve the ionic conductivity[25–27].

In this work, we propose that engineering antiphase boundaries by topotactic transformation is an effective method to optimize the ionic conductivity of epitaxial $Sr_4Fe_6O_{12+\delta}$ films. Using scanning transmission electron microscopy (STEM) combined with electron energy loss (EEL) spectroscopy techniques, different types of APBs with or without the capability of topotactic transformation were identified in this prototype MIECs film[28]. Moreover, the atomic scale dynamic observation of oxygen diffusion and loss during the topotactic transformation at the APBs was carried out, while annealing in an oxidation atmosphere can lead to the formation of interstitial oxygen at these APBs. The ionic conductivities of lithium ions and oxygen ions in perovskite structure calculated by classical molecular dynamics (MD) simulations have been shown a good agreement with the ones observed in experiments[29,30], which indicates MD simulation can be served as an effective way to investigate the ion conductivity result from ion diffusion. So, classical MD simulations were conducted to support the critical role of topotactic transformation on the enhanced ion conductivity at the APBs under elevated temperatures. These results provide further insights into topotactic transformation at APBs and demonstrate that engineering APBs by topotactic transformation can be an effective approach to improving oxide-ion conductivity across boundaries.

## Results and discussion
### Atomic structure of topotactically transformable APBs
The $Sr_4Fe_6O_{13-\delta}$ phase based on perovskite structure displays a layered structure, built up of SrO and $FeO_6/FeO_4/FeO_5$ perovskite-type layers alternately (Fig. 1a). In contrast to common perovskite phases, double $FeO_4$ layers of Fe tetrahedron and double $FeO_5$ layers of Fe pentahedron colored by the pink and blue rectangles in the schematic of Fig. 1a, respectively, can be identified between two perovskite-type $FeO_6$ layers. In $Sr_4Fe_6O_{12}$, double $FeO_4$ layers are stacked together to form a zigzag network connected with the tetrahedron, while $FeO_6$ layers close to double $FeO_4$ layers keep straight in the same plane. Due to the flexible oxygen content in these double layers, topotactic transformation[31] between $Sr_4Fe_6O_{12}$ and $Sr_4Fe_6O_{13}$ can be achieved via annealing under oxidized or reduced conditions. As shown in Fig. 1a, additional oxygen atoms can be stored in double $FeO_4$ layers in $Sr_4Fe_6O_{12}$, leading to a topotactic transformation from $FeO_4$ coordination to $FeO_5$ coordination, as well as the tilting of $FeO_6$ layers close to double $FeO_5$ layers. X-ray diffraction (XRD) analysis shows obvious diffraction patterns belonging to $Sr_4Fe_6O_{13-\delta}$ (See Supplementary Fig. 1). The high-angle annular dark field (HAADF) STEM image in the left side of Fig. 1a shows the atomic structure of as-grown films, which is close to the crystalline structure of $Sr_4Fe_6O_{13}$ with higher oxygen content. After being annealed in $CaH_2$ at 500 °C temperature, we can find that the $FeO_5$ double layers can topotactically transform into $FeO_4$ double layers arranged in a zigzag pattern (shown in the right side of Fig. 1a), as expected in the double $FeO_4$ structure of $Sr_4Fe_6O_{12}$. Besides, when further oxidized under an oxygen atmosphere, additional oxygen atoms can be inserted into this zigzag $FeO_4$ double layer, leading to almost straight double layers and obvious tilting of adjacent $FeO_6$ layers. In particular, the observed capacity of oxygen diffusion at these double iron layers is a platform for oxygen migration in fuel devices[15,18,32–34].

We noticed, as shown in the HAADF STEM image in Fig. 1b, that $Sr_4Fe_6O_{13-\delta}$ films contain multiple APBs such as APB-I, II with clear stacking mismatch along out-of-plane directions. These APBs break the

continuity of double Fe-O layered structure across the in-plane direction, (See Supplementary Fig. 2), which can lead to decreased ionic conductivity[15,25,35]. As shown in Fig. 1c, atomic-scale HAADF STEM images clearly reveal that APBs-I, II between adjacent antiphase domains have the same crystal translation vector around 1/5 [010] of $Sr_4Fe_6O_{12}$, however, with different APB planes. Further investigation of atomic structure and composition across these observed APBs reveals that these APBs are sharp across the boundaries along the out-of-plane direction (See Supplementary Fig. 3), and the Sr/Fe ratio varies across a few atomic positions at these boundaries along the in-plane direction[36–38].

In contrast to these APBs observed in Fig. 1c, the topotactically transformable APBs are identified at the atomic scale, when Sr-O planes disappear and the periodicity of Fe-O planes have been modulated along [010] direction. The HAADF STEM image in Fig. 1d shows four Fe-O atomic layers with a zigzag structure across the APB-III. The stacking sequences of each layer are completely different from the $Sr_4Fe_6O_{13-\delta}$ and $Sr_4Fe_6O_{12}$ phases. More importantly, these zigzag four Fe-O layers at APB-III can be transformed into those with vertical Fe atoms arrangements (see APB-IV in Fig. 1d and Supplementary Fig. 4) via topotactic transformation, as it was found that oxygen atoms can migrate in or out through these four layers. Therefore, these reconstructed four Fe atomic layers at APB-III can act as the oxygen ionic conductive channel which is a bridge of oxygen ionic migration between two adjacent antiphase domains. In comparison, due to the strong bonding between Sr and O, APBs-I, II with Sr-O planes cannot be transformed topotactically, which clearly breaks the continuity of the oxygen ionic conductive channel along in-plane directions.

### Compositional and orbital hybridization of the topotactically transformable APBs
Based on our experimental images of the zigzag region, a supercell model including four zigzag Fe-O layers and the adjacent two perovskite-like layers has been constructed (Fig. 2a). The simulated HAADF STEM images based on the constructed atomic model along two perpendicular zone axes of [001] and [100] are in consistent with the experimental ones as shown in Figs. 2a and S5, respectively, demonstrating the validity of the theoretical model of the supercell. Atomic-resolution element maps of Fe and Sr elements acquired by EDS in Fig. 2b reveal that this unique zigzag structure consists of Fe elements rather than Sr elements. This zigzag structure marked as 2-$FeO_4$ in Fig. 2c has four atomic layers, where two adjacent Fe atomic columns contract in each layer. In particular, the atomic structure of the double Fe atomic layers within these four zigzag Fe-O layers is almost identical to double Fe atomic layers composed of Fe tetrahedral sites in $Sr_4Fe_6O_{12}$, which is marked as 1-$FeO_4$ in Fig. 2c.

The EEL spectra of Fe $L_{2,3}$ and O $K$-edges are acquired from the different layers marked by the different color rectangles on the top panel of Fig. 2c. The Fe-$L_{2,3}$ edges in undistorted octahedral $FeO_6$ sites marked as $FeO_6$ show the slight shoulder feature in the $L_3$ edge, which is attributed to the crystal-field splitting in the octahedral site, besides, the shoulder feature reduces in distorted octahedral $FeO_6$ site due to the reduced crystal splitting effect[28]. but this feature diminishes in other layers, indicating that Fe atoms in these layers have different oxygen coordination in contrast to the undistorted octahedral sites. For O $K$-edges, the pre-peak at 530 eV mainly stems from the electron transitions between occupied $1s$ states to unoccupied $2p$ states of oxygen. Therefore, the relative intensity of the pre-peak (labeled as A in Fig. 2d) located at 530 eV reflects oxygen $2p$ weight in states of predominantly transition-metal $3d$ character of $d$-band filling[39], whereas the main peak (labeled as B and C) is related to hybridization between O $2p$ and $4d$ states of Sr and $4s$ and $4p$ states of Fe[17,40,41]. It should be noted that a remarkable difference between different Fe-O layers can be distinguished on the relative intensity of A and B peaks. Although these peaks of different layers have similar shapes and

energy positions, the pre-peak in 2-FeO$_4$ exhibits much higher intensity than that in FeO$_6$ and FeO$_{4+\delta}$, and even higher than 1-FeO$_4$ layers due to the more interstitial sites than 1-FeO$_4$ layers (See Supplementary Fig. 5). And double Fe-O layers (1-FeO$_4$ and FeO$_{4+\delta}$) also exhibit higher intensity in pre-peak compared with FeO$_6$ layers, which is in good agreement with previous experimental results in epitaxial Sr$_4$Fe$_6$O$_{13-\delta}$ films[42]. Owing to the sensitivity of $d$-band filling for pre-peak in transition oxide[43–45], previous studies showed that the pre-peak can be

used as an indication of estimating the density of hole states. Therefore, the higher intensity of pre-peak in the spectrum indicates a higher concentration of mobile holes in the zigzag region with four Fe-O layers, even compared with double Fe-O layers. Since that previous studies revealed that the electron-hole conductivity of the films can be modulated by tailoring the oxygen content in the double layers, the higher concentration of mobile holes in these four Fe-O layers is beneficial for optimizing the performance of conductivity.

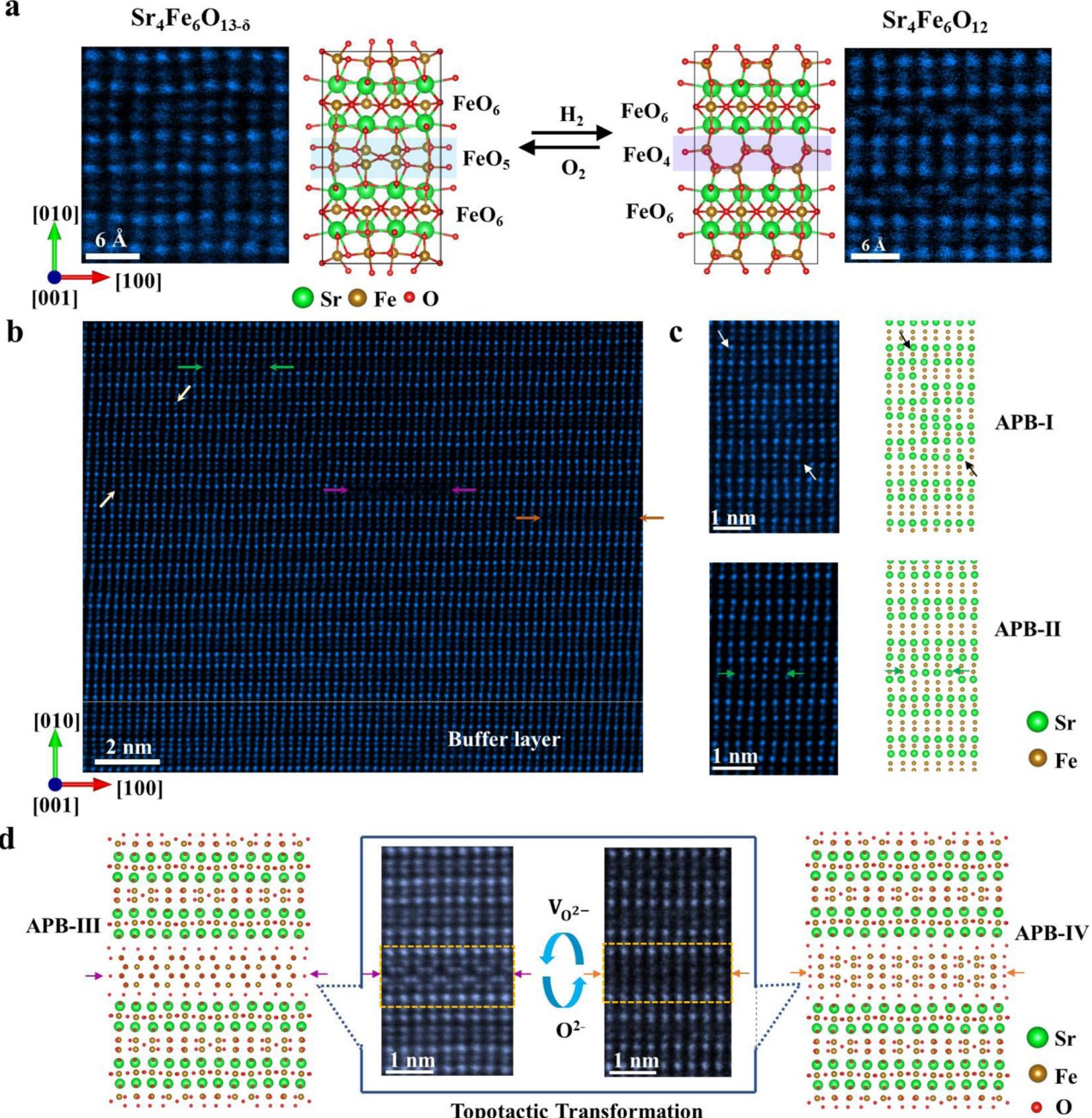

**Fig. 1 | Comparison of the atomic structure of topotactically transformable APBs and topotactically non-transformable APBs. a** The Magnified view of high angle annular dark field scanning transmission electron microscopy- (HAADF STEM) images of Sr$_4$Fe$_6$O$_{12}$ and Sr$_4$Fe$_6$O$_{13}$ with different oxygen content, together with corresponding schematic images of atomic models of Sr$_4$Fe$_6$O$_{12}$ and Sr$_4$Fe$_6$O$_{13}$. By annealing in an oxidized or reduced atmosphere, double Fe layers with different topotactic phases can be transformed reversibly. **b** Experimental atomic resolution HAADF STEM image of Sr$_4$Fe$_6$O$_{13-\delta}$ films along [100] direction. Regions with APBs are highlighted by colored arrows. **c** HAADF STEM images of topotactically non-transformable APB-I and APB-II from (**b**), and the corresponding atomic models. (**d**), HAADF STEM images of topologically transformable APB-III from (**b**), and the corresponding atomic model, where four Fe-O layers are constructed. Atomic structures of four Fe-O layers can be transformed topotactically between APB-III and APB-IV.

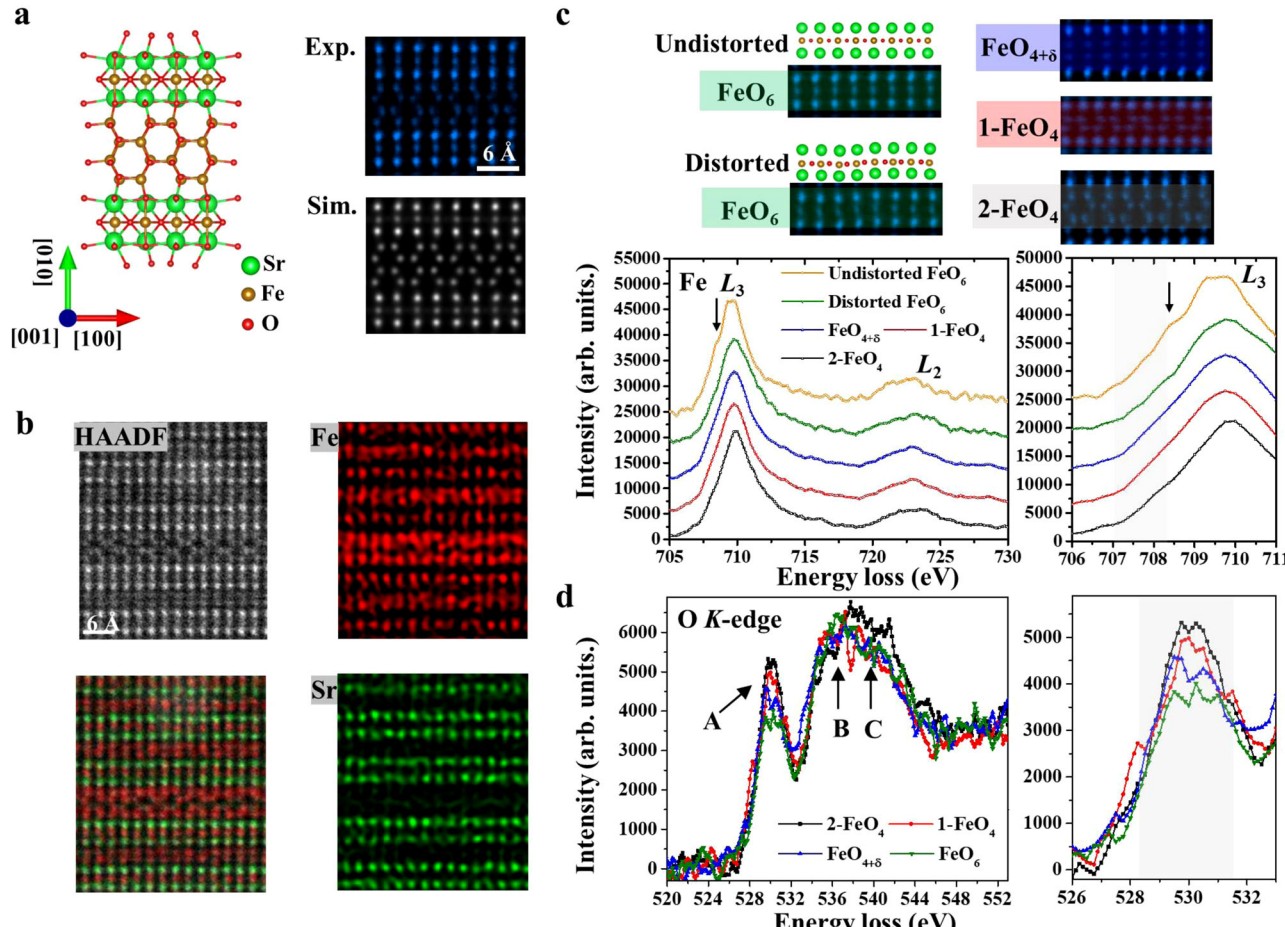

**Fig. 2 | Atomic-scale composition and orbital occupation state analysis of topotactically transformable APB-III. a** Experimental and simulated high angle annular dark field scanning transmission electron microscopy (HAADF STEM) images of four Fe-O layers with a zigzag structure at topotactically transformable APB-III. **b** HAADF STEM image and energy dispersive X-ray spectroscopy (EDXS) maps of Fe and Sr elements at APB-III. **c**, **d** Spectra of Fe $L_{2,3}$ edges and normalized O $K$-edges acquired from the different Fe-O layers as shown in the corresponding HAADF STEM images on top. Distorted $FeO_6$ is referred to as the octahedral Fe-O layer within perovskite layers of $Sr_4Fe_6O_{13-\delta}$. Undistorted $FeO_6$ is referred to as the octahedral Fe-O layer within perovskite layers of APB-III. 1-$FeO_4$ of $Sr_4Fe_6O_{12}$ and 2-$FeO_4$ of APB-III represent the regions consisting of two and four tetrahedral layers, respectively, while $FeO_{4+\delta}$ represents the regions consisting of two pentahedral layers in $Sr_4Fe_6O_{13-\delta}$. Due to the slight variation of oxygen content, this layer may have $FeO_4$ or $FeO_5$, so $FeO_{4+\delta}$ is used here. The feature of the shoulder marked by a black arrow can be resolved in the Fe-$L_3$ edge of undistorted $FeO_6$ layers.

## Interstitial oxygen atoms at the topotactically transformable APBs

Fig. 3a illustrates how additional oxygen atoms can be intercalated into the interstitial sites at the topologically transformable APB-III. The interstitial sites (marked by open circles in Fig. 3a) can be occupied by oxygen atoms. Meanwhile, these interstitial oxygen atoms can also migrate out from these interstitial sites, which is similar to the topotactic transformation between $Sr_4Fe_6O_{12}$ and $Sr_4Fe_6O_{13}$. To gain insight into the oxygen migration across four Fe-O layers at APB-III, annular bright field (ABF) STEM images were acquired along the [001] zone axis to determine the positions of oxygen atoms, and further, understand the different geometry of coordination environment for the Fe element in each layer. As shown in Fig. 3b, atoms with higher contrast in HAADF STEM images represent the Sr atoms, while the atoms between two Sr atomic layers represent octahedral Fe atoms where two adjacent oxygen atoms highlighted by red balls can be resolved. In contrast with octahedral Fe atomic layers, no additional oxygen atoms have been observed in interstitial sites marked by white arrows at the topotactically transformable pristine APBs-III. Besides, along the [100] zone axis (See Supplementary Fig. 6), the HAADF STEM image and EELS analysis reveals a consistent atomic structure and composition with four zigzag layers consisting of Fe and oxygen elements.

Therefore, it can be experimentally confirmed that these four zigzag layers at the topotactically transformable APBs-III exhibit analogous atomic arrangement as double $FeO_4$ layers in $Sr_4Fe_6O_{12}$, where $FeO_4$ tetrahedron connects with each other to form a network.

After being partially oxidized in an oxygen atmosphere (See Supplementary Fig. 7), additional oxygen atoms can be randomly inserted into these interstitial sites at the topotactically transformable APBs, marked by white arrows in Fig. 3c. Due to the limited content of interstitial oxygen atoms, the zigzag feature can still be observed at the topotactically transformable APBs. In addition, oxygen atoms in the interstitial sites can also be resolved in the zigzag region close to the APBs. As shown in the ABF image of Fig. 3d, we find that $FeO_{4+\delta}$ double layers with more oxygen content can be well accommodated to connect with a zigzag region, leading to a complete ionic channel for oxygen atoms across the topologically transformable APBs. These interstitial sites can provide a proper migration path for oxygen atoms across the interstitial sites along the in-plane direction. In contrast to APB-I and APB-II with discontinued oxygen ionic channels at APBs (seen in Fig. 3e), the topotactically transformable APBs in Fig. 3f act as an ideal bridge to connect the ionic migration channel between adjacent domains at the APBs. Besides, it can be found that the distribution of lateral interatomic spacing in the zigzag region has a chessboard-

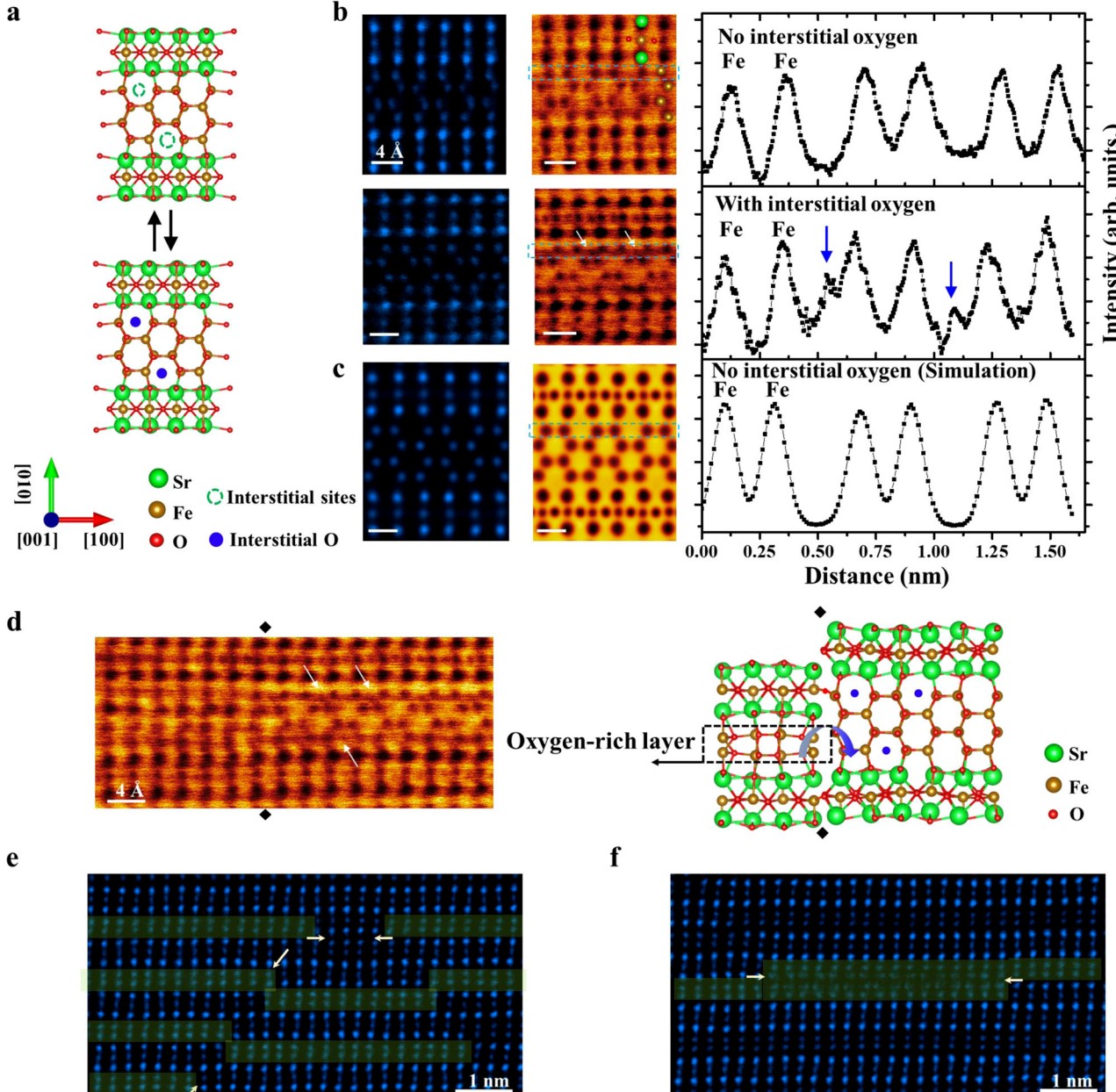

**Fig. 3 | Atomic resolution imaging of interstitial oxygen atoms at the topo-tactically transformable APB. a** Schematic atomic model depicts the interstitial sites for oxygen atoms in the four Fe-O layers with a zigzag structure.
**b** Experimental HAADF STEM and Annular bright-field scanning transmission electron microscopy images of four Fe-O layers with a zigzag structure before and after being partially oxidized, On the left side, the intensities of the line profile extracted from the regions marked by the blue rectangle are plotted. The blue arrows indicate the peaks of interstitial oxygen atoms. Some oxygen atoms can be found in interstitial sites of the zigzag region indicated by white arrows in the ABF

STEM image. **c** Simulated HAADF and ABF STEM image of APBs-III without inter-stitial oxygen atoms. On the left side, the intensity of the line profile extracted from the regions marked by the blue rectangle is plotted. **d** ABF STEM image of the zigzag APBs-III adjacent to FeO$_{4+\delta}$ double layers, clearly revealing interstitial oxygen atoms with significant contrast. The schematic picture is depicted on the right side. **e** The discontinuous in-plane oxygen ionic channels blocked by the APB-I and APB-II marked by white arrows. **f** APB-III shows continued oxygen ionic channel across the APBs, where four Fe-O layers at APB-III can be a bridge to connect the adjacent domains at the APBs.

like feature where two adjacent atomic columns at each layer are measured to be 3.5 Å and 1.5 Å (See Supplementary Fig. 4). While after being fully oxidized in this region, the measured distribution of lateral interatomic spacing in the four Fe-O layers is shown to be uniform with no obvious lattice variation (See Supplementary Fig. 4). Furthermore, EDS and atomic structure analysis indicate that these zigzag regions can be transformed into vertical four layers through topotactic transformation. The atomic arrangements of these vertical four Fe-O layers are similar to that in double Fe-O layers of Sr$_4$Fe$_6$O$_{13}$. (See Supplementary Fig. 4).

## Dynamic observations of topotactic transformations of APBs

To gain further insight into the dynamic evolution of oxygen dif-fusion during topotactic transformation, in-situ STEM observation of atomic-scale topotactic transformation at the four Fe-O layers with vertical atomic arrangements at APB-IV as an initial region was performed under the irradiation of high-energy electrons as a dri-ven force (seen in Fig. 4a). Ref. 46, successfully demonstrate that in-situ observation by using TEM reveals the dynamic transformation of ionic migration. The focused electron probe in the STEM mode led to the migration of oxygen atoms from the interstitial sites at the

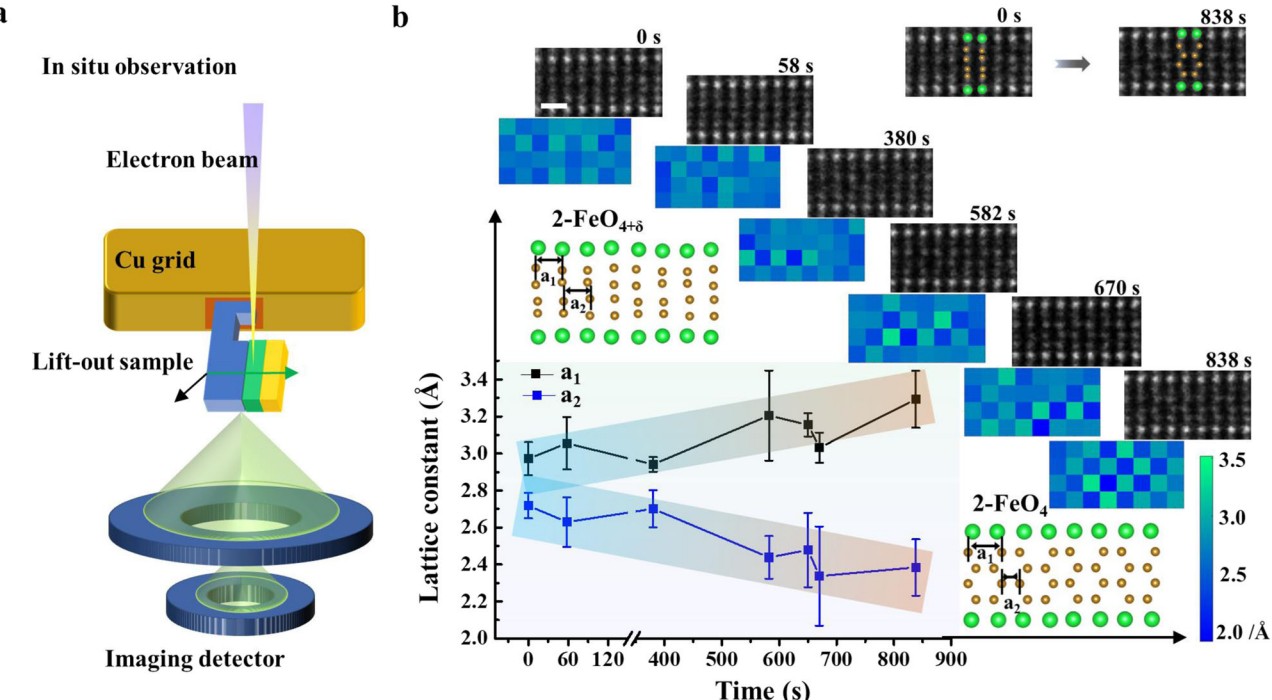

**Fig. 4 | Atomic-scale dynamic observation of topotactic transformation at the topotactically transformable APBs with four Fe-O layers. a** Schematic figure depicts the experimental setup for the in-situ observation of topotactic transformation when subjected to the high-energy electron beam. **b** Quantitative comparison of the distribution of lateral interatomic spacing at the four Fe-O layers at the different stages of the beam exposure experiments. The scale bar is 5 Å. The lateral interatomic spacings defined as $a_1$ and $a_2$ are depicted in the atomic model of the inset. The different atomic models of the four Fe-O layers are superimposed on the HAADF STEM images as shown in the right upper panel. The error bars are defined as the standard deviations of the measured constant in each images.

topotactically transformable APBs (seen in Fig. 4b). It can be found that distribution of lateral interatomic spacing in this region is relatively uniform at the initial stage starting at 0 s, which is ascribed to the saturated oxygen atoms at the interstitial sites. After being irradiated by an electron beam at the different stages (starting from 0 s to 838 s), the vertical four Fe-O layers start to transform topotactically into a zigzag arrangement as shown in the color map, where the original uniform color turns into chessboard-like distribution. Distances between adjacent Fe atomic columns in the four Fe-O layers are quantitatively measured and plotted in Fig. 4, indicating that the $a_1$ constant increases while the $a_2$ constant decreases, respectively.

The more plausible explanation for the topotactic transformation can be attributed to the local stress anisotropy induced by local lattice mismatch and momentum transfer between oxygen atoms and high-energy electrons[47]. The straight four Fe-O layer similar to $Sr_4Fe_6O_{13}$ is under compressive stress along the in-plane direction when growing epitaxially on the oxygen-deficient $Sr_4Fe_6O_{13-x}$, since the in-plane lattice constants are larger than that of oxygen-deficient $Sr_4Fe_6O_{13-\delta}$. The initial fewer oxygen defect due to the 'knock on' effect of beam exposure with low beam current can be considered as the first empty sites. And these interstitial oxygen atoms can migrate through these empty sites across the topotactically transformable APBs due to the energy perturbation of electron beam irradiation when the focused electron beam continuously irradiates the straight four Fe-O layers[26,48]. In contrast, 'knock on' effect induced by electron beam irradiation cannot easily lead to the transformation from $Sr_4Fe_6O_{13}$ to $Sr_4Fe_6O_{12}$ at the same experimental condition (Seen in Supplementary Fig. 8). The less constraint along the out-of-plane direction can enable the four Fe-O structures to expand more by transforming topotactically into the zigzag structure, where oxygen atoms migrate out. Meanwhile, this expansion along the out-of-plane direction can reduce the in-plane strain.

## Enhanced oxygen ionic conductivity at topotactically transformable APBs

Classical MD simulations were performed to study the ionic conductivity of strontium ferrite resulting from the oxygen ions diffusion at the temperature ranging from 300 to 600 K in increments of 50 K (see Materials and Methods for simulation details). Figure 5a, b show the in-plane and out-of-plane components of the resultant ionic conductivity of strontium ferrite with three structures, including $Sr_4Fe_6O_{12}$, $Sr_4Fe_6O_{13}$, and the topotactically transformable APBs with four Fe-O layers respectively. The ionic conductivities of $Sr_4Fe_6O_{13}$ in both directions are negligible compared with those of $Sr_4Fe_6O_{12}$ and the topotactically transformable APBs with four Fe-O layers, indicating $Sr_4Fe_6O_{13}$ is not ionically conductive at the temperature range of 300–600 K. Comparing the ionic conductivities in the in-plane and out-of-plane directions at the desired working temperature, the in-plane ionic conductivity is much greater than the out-of-plane one, suggesting that the in-plane ionic conductivity is the major component of overall ionic conductivity. The in-plane ionic conductivity of the topotactically transformable APBs with four Fe-O layers increases monotonically with the temperature, exhibiting the highest ionic conductivity of $7.6 \times 10^{-5}$ S/cm at 550 K, while the temperature dependence of in-plane ionic conductivity of $Sr_4Fe_6O_{12}$ is not obvious and the maximum ionic conductivity exists at 400 K. The superior in-plane ionic conductivities of the topotactically transformable APBs with four Fe-O layers can be observed compared with those of the $Sr_4Fe_6O_{12}$ at a temperature higher than 500 K.

To probe the origins of the discrepancy in the ionic conductivities of different structures at different temperatures, the oxygen ion densities at 400 and 550 K of different structures are shown in Fig. 5c–e. The oxygen ion densities demonstrate that the migration of oxygen ions in the out-of-plane direction is hindered by the strontium layers, causing the low out-of-plane ionic conductivity. The localized displacements of oxygen ions in $Sr_4Fe_6O_{13}$ at both 400 and 550 K (Fig. 5d)

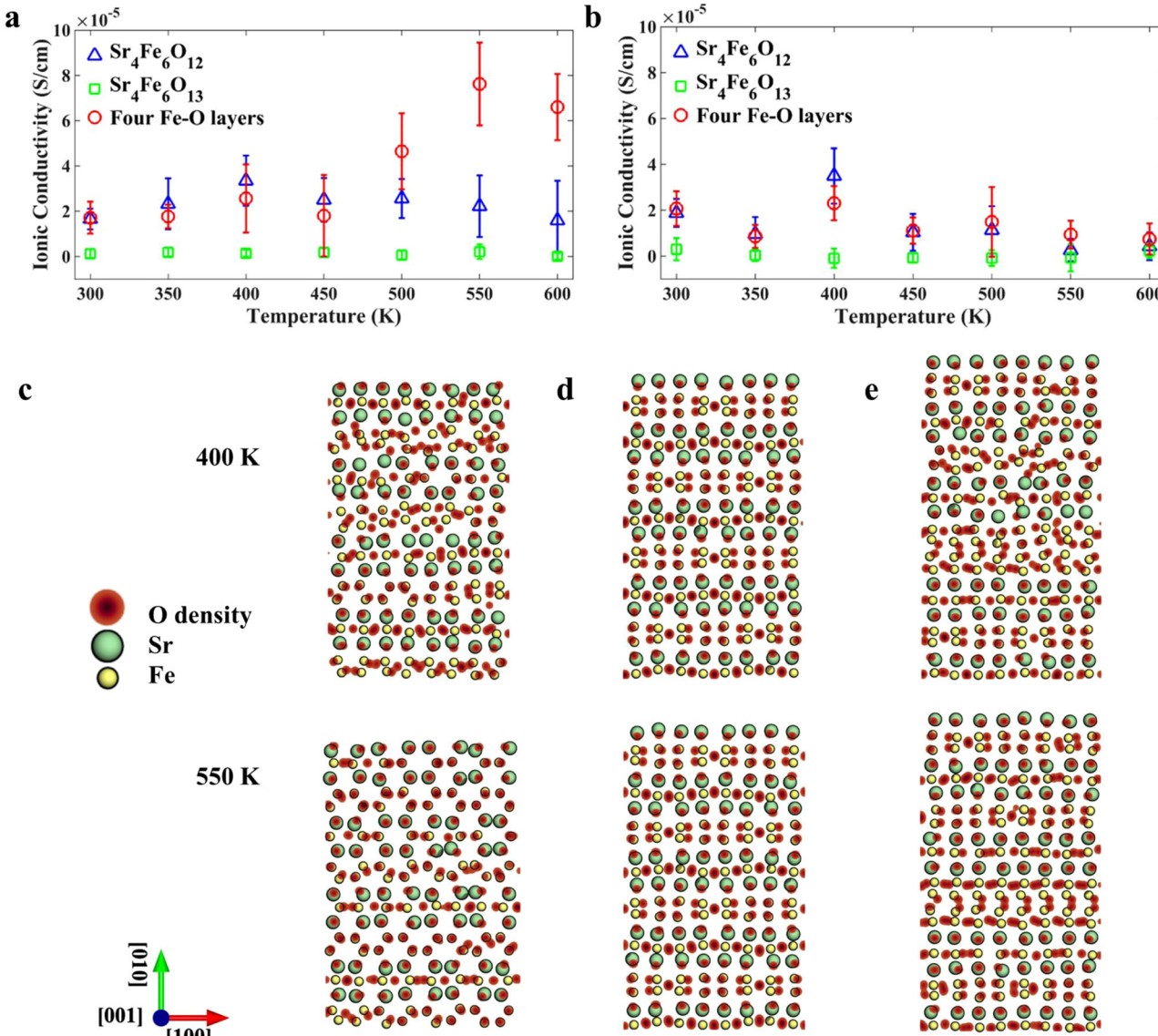

**Fig. 5 | Temperature dependence of ionic conductivity of $Sr_4Fe_6O_{12}$, $Sr_4Fe_6O_{13}$, and the topotactically transformable APBs-III with four Fe-O layers structures.** The in-plane (**a**) and out-of-plane (**b**) components of ionic conductivity of strontium ferrite as a function of temperature with different structures ($Sr_4Fe_6O_{12}$, $Sr_4Fe_6O_{13}$, and the topotactically transformable APBs-III with four Fe-O layers structures). The error bars are defined as the standard deviations of calculations. The oxygen ion densities in $Sr_4Fe_6O_{12}$ (**c**), $Sr_4Fe_6O_{13}$ (**d**), and the topotactically transformable APBs-III with four Fe-O layers structures (**e**), at 400 and 550 K.

confirm that the oxygen ions in $Sr_4Fe_6O_{13}$ are very stable and can only move locally. The restrained movements of oxygen ions result in negligible ionic conductivity. On the other hand, the elongated oxygen ion densities in $Sr_4Fe_6O_{12}$ and the topotactically transformable APBs with four Fe-O layers (Fig. 5c, e) along the in-plane direction reflect that the substantial in-plane motions of oxygen ions. The high in-plane mobility of oxygen ions gives rise to enhanced in-plane ionic conductivity. The elongated oxygen ion densities of four Fe-O layers and $Sr_4Fe_6O_{12}$ reveal that the topotactically transformable APBs with four Fe-O layers can renovate the in-plane ionic conductivity at discontinued boundaries (APB-I and APB-II). The broader distribution of oxygen ion density along the in-plane direction of the topotactically transformable APBs with four Fe-O layers at 550 K than that at 400 K (Fig. 5e) indicates the increasing in-plane ionic conductivity with temperature is mainly attributed to the thermally activated process. For $Sr_4Fe_6O_{12}$, the structural difference at 400 and 550 K can be observed (Fig. 5c), which is consistent with the neutron powder diffraction data that $Sr_4Fe_6O_{12}$ is unstable when temperature is near

600 K[47]. The structural change at 550 K makes the oxygen ions difficult to move in the in-plane direction, which can be seen from the oxygen density in Fig. 5c, leading to the degraded in-plane ionic conductivity comparing with that at 400 K. The simulation results suggest that the introduction of four Fe-O layers increases the pathways for the migration of oxygen ions, contributing to the enhancement in ionic conductivities at the temperature range of 500–600 K.

In summary, engineering antiphase boundaries with the capability of topotactic transformation has been proposed and demonstrated to enhance ionic conductivity in MIECs. In the example of epitaxial $Sr_4Fe_6O_{13-\delta}$ films, APBs without the capability of topotactic transformation hinder the ionic migration since they break the in-plane continuity of layered structure as well as anion exchange. In contrast, ex-situ and in-situ atomic-scale imaging experiments, together with MD simulations uncover that Fe-O tetrahedrons at topologically transformable APBs are constructed together with interstitial sites which can serve as oxygen migration channels with enhanced ionic conductivity across the boundaries at elevated temperature, as well as the

bridge between two adjacent antiphase domains with in-plane discontinuity. These synergistic investigations pave a new path for the structure design of ionic conductors by utilizing topotactic transformation as an approach of defect engineering.

# Methods

## Thin-film growth

The high-quality $Sr_4Fe_6O_{13-\delta}$ films were epitaxially grown on (001) $LaAlO_3$ single crystal substrates by the pulsed laser deposition method with a KrF excimer laser ($\lambda = 248$ nm) at a repetition rate of 10 Hz and a laser fluence of 2–3 J/cm. The substrates were maintained at the optimized temperature of 750 °C with oxygen partial pressure $P(O_2)$ of 2.8–3 Pa during deposition. The deposition time for film growth was 30 min. The samples were annealed for 30 min to be fully oxidized at the same growth temperature and oxygen partial pressure. After deposition, the samples were cooled down at 5 °C/min under the same oxygen flow. The crystalline quality was characterized by high-resolution XRD using a Rigaku Smartlab X-ray diffractometer. The reduction and re-oxidation processes were completed using TEM samples directly. The re-oxidation process was achieved by annealing the TEM sample in a high oxygen partial pressure environment ($P(O_2)$ of ~50 Pa) at 400 °C for 4 h. The warming and cooling rates were both 10 °C/min. The reduction process was achieved by annealing the TEM sample at 400 °C for 4 h. The TEM sample was placed in the crucible, which was filled with $CaH_2$ powder. The crucible was covered by aluminum foil. In-plane AC impedance measurements were carried out by sputtering platinum interdigitated electrode (150 μm space, and 2000 μm arm length) onto the film surface. The AC impedance measurements were performed in the frequency range of 20 Hz–10 MHz by an HP 4990 A Precision Impedance Analyzer with 50 mV voltage. Variable temperature measurements between 300 K and 673 K were realized by a probe station equipped with a hot plate.

## STEM characterization and analysis

The cross-sectional specimens for STEM observation along different zone axes were prepared using a Zeiss Auriga focused ion beam (FIB) system. To minimize the beam damage in FIB, an approximately 5 nm thick Au layer and a 2 μm thick polycrystalline Pt layer was deposited on the surface of films before the ion thinning process. The cross-sectional TEM specimens were firstly thinned by $Ga^+$ion beams and then polished by $Ar^+$ ion beam (Nanomill) to minimize the surface damage and amorphous layers induced by previous $Ga^+$ ion milling, with an acceleration voltage of 500 V, 100 mA for 30 min. Atomic-resolution STEM images were acquired in a FEI Titan Cubed Themis G2 300 operated at 300 kV equipped with a high-brightness Schottky field emission gun and monochromator, a probe aberration corrector to provide a spatial resolution better than 0.6 Å in the STEM mode, and energy dispersive X-ray spectroscopy and Gatan imaging filter quantum energy filters for EEL spectroscopy analysis. Z contrast imaging was conducted in HAADF STEM mode with a probe semi-convergence angle of 25 mrad and an inner collection angle of approximately 80 mrad and an outer collection angle of approximately 240 mrad. HAADF imaging was combined with EDS for element mapping. The collection semi-angle for ABF STEM imaging was from 11 to 22.4 mrad. The energy drifts of all the spectra were corrected by using zero-loss spectrum acquired simultaneously, while the background was subtracted from the spectrum image to show the delicate near-edge fine structures of each element. The beam current of 10 pA at STEM mode was employed to achieve the dynamic observation at tt-APBs. Atomic column positions in the HAADF STEM images were determined using the two-dimensional (2D) Gaussian fitting method embedded in Mac-TEMPAS. Next, the integrated intensities of atomic columns were extracted over the defined ranges of the fitted 2D Gaussian peak centered at each peak position. A Matlab code was used to precisely determine the center of mass of the projected atomic columns in the

HAADF STEM images to derive the local lattice parameters. The image simulations were performed using the Multi-slice method by the Dr. Probe software[49].

## Molecular dynamics simulation

The initial configurations of atomic models were created by expanding the unit cell obtained from density function theory calculation. The atomic models of strontium ferrite consisted of 22,528 ions (4096 Sr, 6144 Fe, and 12,288 O), 23,552 ions (4096 Sr, 6144 Fe, and 13,312 O), and 25,344 ions (4096 Sr, 7168 Fe, and 14,080 O) with the initial dimensions of $89.6 \times 77.5 \times 44.8$, $88.5 \times 79.4 \times 43.3$, and $89.6 \times 84.0 \times 44.8$ Å$^3$ for $Sr_4Fe_6O_{12}$[50], $Sr_4Fe_6O_{13}$, and the topotactical transformable APBs with four Fe-O layers structures, respectively.

The interactions between ions were based on the Born-like model, including the long-range Coulombic and short-range Buckingham interactions.

$$\Phi_{ij} = \sum_{j>i} \sum \left[ \frac{q_i q_j}{4\pi\varepsilon_0 r_{ij}} + A_{ij} \exp\left(\frac{-r_{ij}}{\rho_{ij}}\right) - \left(\frac{C_{ij}}{r_{ij}^6}\right) \right] \quad (1)$$

where $q_i$ and $q_j$ are the charges of ions $i$ and $j$, respectively, $r_{ij}$ is the distance between ions $i$ and $j$, and $A_{ij}$, $\rho_{ij}$, and $C_{ij}$ are Buckingham potential parameters, which were taken from the previous study on strontium ferrite materials[33]. The electronic polarizability of the oxygen ions was considered by the shell model[51]. The oxygen ion was represented as a shell connecting to a core by a harmonic spring to mimic the electron shell of an ion. The charges of the shell and core were $-2.21$ $e$ and $0.21$ $e$, respectively, and the force constant of the spring was 21.29 eV/Å$^2$. The mass ratio of the shell to the core was set to 0.01. The Coulomb interaction was treated using the particle-particle particle-mesh Ewald summation method[52] and the cutoff distance of Buckingham potential was set to 10 Å. The periodic boundary conditions were applied to all directions. The equations of motion were integrated with a time step of 0.5 fs. All the simulations were carried out using large-scale atomic/molecular massively parallel simulator (LAMMPS) package[53] and visualized with the software PyMOL[54].

The Nernst-Einstein equation was implemented to calculate the ionic conductivity, $\sigma$:

$$\sigma = \frac{Nq^2D}{Vk_BT} \quad (2)$$

where $N$, $q$, and $D$ are the number, point charge, and self-diffusion coefficient of ion, respectively, $V$ and $T$ are the volume and temperature of the simulation system, and $k_B$ is the Boltzmann constant. The self-diffusion coefficient was obtained from the mean squared displacement according to the Einstein relation:

$$D = \frac{1}{6t} \left\langle \left[ r_i(t+t_0) - r_i(t_0) \right]^2 \right\rangle \quad (3)$$

where $r_i(t+t_0)$ and $r_i(t)$ are the position of $i^{th}$ ion at time $t+t_0$ and $t_0$, respectively, <...> denotes the ensemble average. The linear fitting of the plot of the mean squared displacement with time between 50 and 200 ps was used to calculate the self-diffusion coefficient. The diffusion of oxygen ions is more significant than that of strontium and iron ions, so only the diffusion of oxygen ions was considered to calculate ionic conductivity. To investigate the anisotropic ionic conductivity, the ionic conductivity was decomposed into the in-plane component ([100] zone axis) and the out-of-plane component ([010] zone axis).

The system was initially relaxed in the isothermal-isobaric (NPT) ensemble at the pressure of 0 bar and the target temperature for at least 1 ns until the equilibrium lattice structure was achieved. The

follow-up equilibrium run in the canonical (NVT) ensemble was performed for 0.5 ns to ensure the system temperature. Finally, an overall 2.5 ns production run was conducted in the microcanonical (NVE) ensemble and the ionic conductivity was calculated every 0.5 ns.

## Data availability

All data supporting the results of this study are available in the manuscript or the supplementary information. All related data are accessible in figshare[55].

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

## Acknowledgements

This work was financially supported by the Chinese National Natural Science Foundation (Basic Science Center Project of NSFC under grant No. 52388201 (J.Z., P.Y.), 11834009 (J.Z.)), and supported by the Basic and Applied Basic Research Major Programme of Guangdong Province, China (Grant No. 2021B0301030003) (J.Z.), and Jihua Laboratory (Project No. X210141TL210) (J.Z.). This work made use of the resources of the National Center for Electron Microscopy in Beijing and the Tsinghua National Laboratory for Information Science and Technology. X.Y. Z is grateful for the financial supports from National Natural Science Foundation of China (52171014 (X.Y.Z.), 52011530124 (X.Y.Z.), 52025024 (P.Y.)), Science, Technology and Innovation Commission of Shenzhen Municipality (SGDX20210823104200001, JCYJ20210324134402007, HZQB-KCZYB-2020031) (X.Y.Z.), the Sino-German Mobility Programme by the Sino-German Center for Research Promotion (M-0265) (X.Y.Z.), Innovation and Technology Fund (ITS/365/21) (X.Y.Z.), Science and Technology Department of Sichuan Province (2021YFSY0016) (X.Y.Z.), the Research Grants Council of Hong Kong Special Administrative Region, China (Project No. E-CityU101/20, CityU 11302121, CityU 11309822, G-CityU102/20) (X.Y.Z.), the European Research Council (Grant No. 856538, project "3D MAGiC") (X.Y.Z.), CityU Strategic Interdisciplinary Research Grant (7020016, 7020043) (X.Y.Z.), the City University of Hong Kong (Projects no. 9610484, 9680291, 9678288, 9610607) (X.Y.Z.), the City University of Hong Kong Shenzhen Research Institute and City University of Hong Kong Chengdu Research Institute. We thank Dr. Jinlian Lu and Dr. Xiang Chen for the fruitful discussions.

## Author contributions

K.X. carried out the TEM specimen preparation, TEM data acquisition, and TEM-related simulation. C.R.H. conducted the impedance measurements of the conductivity of prepared films. K.X., W.L.S, X.Y.Z and J.Z. performed data processing and wrote the manuscript. K.X., Y.S.W. and P.Y. grew samples and contributed to writing the manuscript. P.Y. and X.Y.Z. contributed to the discussion section of the manuscript. S.W.H. carried out the MD calculation work and analyzed the calculation results. K.X. and X.Y.Z. proposed ways to analyze the experimental results and theoretical calculation from the point of view of topotactically transformable defect engineering. The manuscript was written through the contributions of all authors. All authors have approved the final version of the manuscript.

## Competing interests

The authors declare no competing interest.
