## [Peer Review File · Nature Communications]

Reviewers' Comments:

Reviewer #1:

Remarks to the Author:

The work from Kun Xu et al discovered a type of topoactically transformable antiphase boundary (tt-APB) in $\text{Sr}_4\text{Fe}_6\text{O}_{13-\delta}$ layers, identified the atomic structure in detail, observed the interstitial oxygen atoms through in-situ STEM and did MD simulation to support the conclusion that tt-APB can increase oxygen conductivity by forming anion diffusion channels. The finding that the tt-APB can enhance the ionic conductivity is interesting and of importance for the defect engineering study. The electron microscopic characterizations and analysis of the defects atomic structure and dynamics observation of the transformation process are solid. The paper is organized and written properly. However, the conclusion that the tt-APB can enhance the ionic conductivity lacks more straightforward experimental evidence like ionic conductivity measurement of $\text{Sr}_4\text{Fe}_6\text{O}_{13-\delta}$ sample that contains tt-APB. Hence, I recommend major revision of this work before further consideration for publication on Nature Communications. Following are questions for the authors to address:

(1) For the EEL spectra (Figure 2c) of Fe L_{2,3} edges in octahedral FeO₆ sites show slight shoulder feature, which is not obvious. And there is similar level of shoulder feature for 1-FeO₄, 2-FeO₄.

The authors made too strong conclusion based on eye-observation. Can the authors explain this?

(2) In Figure 2d, what is the exact intensity ratio of peak A and Peak B/C for each sample? Please quantify it and check whether it fit the conclusion that the authors made based on this piece of data.

(3) How frequent can tt-APB be observed for $\text{Sr}_4\text{Fe}_6\text{O}_{13-\delta}$ film? Can it be controlled?

(4) Although interstitial oxygen atoms could form in tt-APB, and MD simulation support the enhanced ionic conductivity with the existence of tt-APB, there is no straight experimental evidence for this conclusion, which seems to be hasty. Can the authors provide directly measured ionic conductivity data for the films?

(5) Some minor points:

Please add the sample names in Figure 1a for left and right side.

In Figure 2c and 2d, the vertical axis needs to be labeled with exact numbers.

Reviewer #2:

Remarks to the Author:

The manuscript describes the atomic-scale observation of topotactically transformable antiphase boundaries of strontium ferrite oxide epitaxial thin films and the molecular dynamics simulations of the temperature dependence of ionic conductivity. The detailed observations and supporting simulation results are impressive. I agree with the claim that this study will open a new path of defect engineering strategies for improving ionic transportation by utilizing topotactic transformation. However, in the current version, there are some points that need to be considered, see below:

1. "antiphase boundaries (APBs)" is too technical a term to understand. It is advisable to include an explanation of "APBs" in the Introduction section.

2. Fig. 1a contains the magnified views of the TEM images, but the caption does not. The caption should be correctly described.

3. One of the most important claims of this paper are that the zigzag four Fe-O layers at APB-III can be transformed before and after fully being oxidized, as shown in Fig. 1d and Fig. S4. However, the reduction and re-oxidation methods are not described in the Methods section. It is very important for this paper to specify these methods.

4. Why is the in-plane ionic conductivity greatest at 550 K? Why does ionic conductivity decrease at higher temperatures? There should be more discussion about the relationship between temperature and ionic conductivity.

5. In Fig. S1, the caption should be changed because the coherency of the film cannot be determined from this XRD pattern alone. Note that the thin-film coherency is fully evident from the TEM images.

6. In Fig. S2, the difference between (a,b) and (c,d) is not clear from the caption. In addition, the caption should state the substance name.

7. In Fig. S4, the caption should state the substance name.

8. Minor points and typos:

Line 85: "High" change to "high"

Lines 244, 255, 257: "topotacitcally" change to "topotactically"

Line 306: Subscript of chemical formula

Reviewer #3:

Remarks to the Author:

The paper describes details of the topotactic defects in Sr-Fe-O oxides and discusses their effects on the ionic conductivity. Various type of the local defects, termed as being of antiphase-boundary type (APB), have been successfully identified based on atomic-resolution microscopy analysis, and the occurrence of such variations are reasonably related with the local oxygen contents, as investigated by microbeam spectroscopy. Overall the manuscript is well organized, but I find that quantitative estimations are lacking in local oxygen imaging as well as spectroscopy analysis. In addition, there could be a problematic point on the supposed oxygen behavior during in-situ beam-irradiating observations. Because of these, I cannot recommend the present manuscript for publication in Nature Communications. Detailed comments are described below.

1. Quantitative analysis is needed for both oxygen atom imaging and local oxygen content.

A series of dark-field images of cation sites are convincing and well supported by simulations. On the other hand, bright-field imaging of oxygen atoms is not clear enough and hence not convincing, in particular for those located at interstitial sites (e.g, Fig. 3d). These should be shown in a more quantitative manner, e.g., intensity profile comparisons with the simulations. Also, the explanations on local oxygen contents by EELS profiles (Fig. 2c, d) are too qualitative. They should be evaluated quantitatively, by showing comparisons with the simulations and/or Fe/O ratio derived from each EELS profile.

2. Oxygen atom behaviors during the in-situ electron-beam radiations.

The present in-situ beam-radiation experiments are aimed at promoting local oxygen diffusion. However, as is well known, irradiation with high-energy electron beams (acceleration voltage ~ 100 kV) is most likely to cause so-called 'knock-on' effects, which extract oxygen atoms from the specimen. This seems to be consistent with the results in Fig. 4b; change from $\text{FeO}_{4+\delta}$ to FeO_4 (i.e., oxygen deficient reaction). To prove that the observed structural change is indeed caused by the in-plane oxygen diffusion, the authors should show the oxygen increase at the neighboring regions, and confirm there is no change of total oxygen contents during the structure change including the neighbor regions (i.e., local decrease/increase oxygen atoms are indeed in a trade-off relation across the certain range).

Reviewer #4:

Remarks to the Author:

This manuscript presents the first atomic-resolution visualization of topotactical transformation occurred at antiphase boundary (APB) in epitaxial $\text{Sr}_4\text{Fe}_6\text{O}_{12+\delta}$ films for using as mixed ionic-electronic conductors (MIECs). Using a high-resolution scanning transmission electron microscopy (STEM) combined with electron energy loss spectroscopy (EELS), the authors showed the high-quality high-angle annular dark field (HAADF) STEM and annular bright field (ABF) STEM images of

various antiphase boundaries with clearly identifying Fe and oxygen individual atoms. Because probing the oxygen migration channels for optimizing the ionic conductivity has been very important and challenging issue in the solid oxide fuel cell field, this work is so unique and interesting with some very excellent STEM imaging presented. In addition, the authors suggested special type of APBs with the capability of topotactic transformation showing dynamic changes of oxygen ions while annealing in an oxidation condition and irradiating an electron beam during STEM observation. This work is very interesting finding and the results are so clear. The manuscript is well written and organized. Therefore, I recommend this manuscript to be published in this journal after addressing the minor comments.

1. The authors showed the migration and diffusion of oxygen ions during oxidation (Fig. 3) and e-beam irradiation (Fig. 4). Because the STEM results can be dependent on the annealing and e-beam irradiation conditions, the detailed experimental procedures should be described in METHOD part for the reader.

Responses to comments

We gratefully thank the editor and all reviewers for the valuable comments and constructive suggestions, which substantially helped us improve the quality and clarity of the revised manuscript. All the revisions are highlighted in yellow in the revised manuscript and supplementary information. The point-to-point responses to all the comments of the reviewers are listed in detail as follows.

REVIEWER COMMENTS

Reviewer #1 (Remarks to the Author):

The work from Kun Xu et al discovered a type of topoactically transformable antiphase boundary (tt-APB) in $\text{Sr}_4\text{Fe}_6\text{O}_{13-\delta}$ layers, identified the atomic structure in detail, observed the interstitial oxygen atoms through in-situ STEM and did MD simulation to support the conclusion that tt-APB can increase oxygen conductivity by forming anion diffusion channels. The finding that the tt-APB can enhance the ionic conductivity is interesting and of importance for the defect engineering study. The electron microscopic characterizations and analysis of the defects atomic structure and dynamics observation of the transformation process are solid. The paper is organized and written properly. However, the conclusion that the tt-APB can enhance the ionic conductivity lacks more straightforward experimental evidence like ionic conductivity measurement of $\text{Sr}_4\text{Fe}_6\text{O}_{13-\delta}$ sample that contains tt-APB. Hence, I recommend major revision of this work before further consideration for publication on Nature Communications. Following are questions for the authors to address:

Response: Thanks for the reviewer's positive comments on this manuscript. We gratefully thank to referee for the valuable comments and constructive suggestions, which helped us improve the quality and clarity of the revised manuscript.

(1) For the EEL spectra (Figure 2c) of Fe L_{2,3}, edges in octahedral FeO₆ sites show slight shoulder feature, which is not obvious. And there is similar level of shoulder feature for 1-FeO₄, 2-FeO₄. The authors made too strong conclusion based on eye-observation. Can the authors explain this?

Response: Thanks for pointing this out. We are sorry for our misleading interpretation. We carefully reconsider the interpretation of this slight feature at the near-edge fine structure of the EEL spectra of FeO₆ sites. We totally agree with the referee's opinion that this feature is not obvious. This slight feature is not obvious due to the fact that these FeO₆ sites in $\text{Sr}_4\text{Fe}_6\text{O}_{13-\delta}$ is distorted when compared to standard FeO₆ octahedrons in ferrite oxide (Ref: [1]Tian, H., et al., G. V. (2014). *Advanced Materials*, 26(3), 461-465; [2] Rossell, M.D., et al., 2013. *ACS nano*, 7(4), pp.3078-3085.). As shown in the HAADF STEM image of FeO₆ in the updated Figure 2c, when the FeO₆ octahedron is distorted, the crystal field splitting energy in FeO₆ decreases so that the feature as a shoulder at the Fe-L₃ edge will not be obvious when compared with undistorted FeO₆ octahedron with larger crystal field splitting effect [Ref: [1]Tian, H., et al., G. V. (2014). *Advanced Materials*, 26(3)]. To avoid the misinterpretation of this slight feature, we

provide the additional ABF image to show that the FeO_6 layer exhibits an obvious octahedral structure, but the FeO_6 is tilted and distorted along the in-plane direction as shown in Figure R1. Besides, we have also provided the additional Fe- $L_{2,3}$ edge EEL spectra of standard FeO_6 without distortion in APB-III. It can be found that there is an obvious shoulder in the Fe- L_3 edge, which can well support our conclusion in this manuscript. To make this interpretation clear and robust, Fe- $L_{2,3}$ edge EELS of standard FeO_6 without distortion in APB-III has been added in Figure 2c.

Figure R1. Comparison of Fe- $L_{2,3}$ edge EELS of FeO_6 with/without distortion. (a) HAADF STEM image of FeO_6 without distortion in perovskite layers of APB-III. (b) ABF, HAADF STEM image of FeO_6 with distortion in perovskite layers of $\text{Sr}_4\text{Fe}_6\text{O}_{13-\delta}$. (c) Comparison of Fe- L_3 EELS of FeO polyhedron. The black arrow indicates the shoulder feature in the Fe- L_3 edge.

Revision to question 1:

The following information is revised in Figure 2c.

Fig. 2. Atomic-scale composition and orbital occupation state analysis of topotactically transformable APB-III. **a**, Experimental and simulated high angle annular dark field scanning transmission electron microscopy (HAADF STEM) images of four Fe-O layers with a zigzag structure at topotactically transformable APB-III. **b**, HAADF STEM image and energy dispersive X-ray spectroscopy (EDXS) maps of Fe and Sr elements at APB-III. **c, d**, Spectra of Fe $L_{2,3}$ edges and normalized O K -edges acquired from the different Fe-O layers as shown in the corresponding HAADF STEM images on top. Distorted FeO_6 is referred to the octahedral Fe-O layer within perovskite layers of $\text{Sr}_4\text{Fe}_6\text{O}_{13-\delta}$. Undistorted FeO_6 is referred to the octahedral Fe-O layer within perovskite layers of APB-III. 1- FeO_4 of $\text{Sr}_4\text{Fe}_6\text{O}_{12}$ and 2- FeO_4 of APB-III represent the regions consisting of two and four tetrahedral layers, respectively, while $\text{FeO}_{4+\delta}$ represents the regions consisting of two pentahedral layers in $\text{Sr}_4\text{Fe}_6\text{O}_{13-\delta}$. Due to the slight variation of oxygen content, this layer may have FeO_4 or FeO_5 , so $\text{FeO}_{4+\delta}$ is used here. The feature of the shoulder marked by a black arrow can be resolved in the Fe L_3 edge of undistorted FeO_6 layers.

The following information is revised in the Results and Discussion parts, line 173.

‘The Fe- $L_{2,3}$ edges in undistorted octahedral FeO_6 sites marked as undistorted FeO_6 show the slight shoulder feature in the L_3 edge, which is attributed to the crystal-field splitting in the octahedral site, besides, the shoulder feature is not obvious in distorted octahedral FeO_6 site due to the reduced crystal splitting effect.²⁸ However, this feature diminishes in other layers, indicating that Fe atoms in these layers have different oxygen coordination in contrast to the undistorted octahedral sites.’

(2) In Figure 2d, what is the exact intensity ratio of peak A and Peak B/C for each sample? Please quantify it and check whether it fit the conclusion that the authors made based on this piece of data.

Response: Thanks for pointing this out. To address this comment of the reviewer we have provided additional quantitative analysis on the oxygen K -edge as the referee suggested. As shown in Figure S5, The ratios of integrated intensities of peak A and peak B are listed in the different FeO layers. It can be found that the I_A/I_B in 2- FeO_4 of tt-APBs is larger than other layers, which is consistent with the interpretation in the manuscript. the pre-peak in 2- FeO_4 exhibits much higher intensity than that in FeO_6 and $\text{FeO}_{4+\delta}$, and even higher than 1- FeO_4 layers due to the more interstitial sites than 1- FeO_4 layers. And double Fe-O layers (1- FeO_4 and $\text{FeO}_{4+\delta}$) also exhibit higher intensity in pre-peak compared with FeO_6 layers. To support our interpretation, the additional quantitative analysis on the oxygen K -edge is provided in Supplementary Materials.

Revision to question 2:

The following information is added in Fig. S5 of Supplementary Materials.

Fig. S5. The quantitative analysis of the ratios of integrated intensities of peak A and peak B.

The following information is added in the Results and Discussions part, line 187.

‘It should be noted that a remarkable difference between different Fe-O layers can be distinguished on the relative intensity of A and B peaks. Although these peaks of different layers have similar shapes and energy positions, the pre-peak in 2-FeO₄ exhibits much higher intensity than that in FeO₆ and FeO_{4+δ}, and even higher than 1-FeO₄ layers due to the more interstitial sites than 1-FeO₄ layers (Supplementary Fig. S5).’

(3) How frequent can tt-APB be observed for Sr₄Fe₆O_{13-δ} film? Can it be controlled?

Response: Thanks for the referee’s good question. We have provided additional HAADF STEM images with different magnifications as shown in Figure R3. tt-APBs can be seen in large amounts of regions of Sr₄Fe₆O_{13-δ} films, although the quantities of tt-APBs are less than non-topotactically transformable APBs. The formation of tt-APBs could be modulated by epitaxial strains and Sr/Fe ratio. In future work, these growth conditions can be taken into consideration to optimize and control the formation of tt-APBs.

Figure R2. HAADF STEM images of prepared Sr₄Fe₆O_{13-δ} films. White rectangles indicate the tt-APBs.

Revision to question 3:

The following information is added in Fig. S1 of Supplementary Materials.

Fig. S1. X-ray Diffraction characterization of as-grown $\text{Sr}_4\text{Fe}_6\text{O}_{13-\delta}$ films (a) X-ray θ - 2θ diffraction patterns of prepared epitaxial film on a LaAlO_3 substrate, where no additional peaks are observed. (b) HAADF-STEM images of different regions in prepared $\text{Sr}_4\text{Fe}_6\text{O}_{13-\delta}$ films. White rectangles indicate the tt-APBs.

(4) Although interstitial oxygen atoms could form in tt-APB, and MD simulation support the enhanced ionic conductivity with the existence of tt-APB, there is no straight experimental evidence for this conclusion, which seems to be hasty. Can the authors provide directly measured ionic conductivity data for the films?

Response: Thanks for this constructive comment. Yes, the measured total conductivity data for the thin films is shown in Figure R3, which is well consistent with the previous result (Ref: [1] *Advanced Functional Materials* 18.5 (2008): 785-793.). In-plane AC impedance measurements were carried out by sputtering platinum interdigitated electrode (150 μm space, and 2000 μm arm length) onto the film surface. The AC impedance measurements were performed in the frequency range of 20 Hz - 10 MHz by an HP 4990A Precision Impedance Analyser with 50 mV voltage. Variable temperature measurements between 300 K and 673 K were realized by a probe station equipped with a hot plate. Figure R3 shows the total conductivity dependence with temperature for films, consistent with the total conductivity in $\text{Sr}_4\text{Fe}_6\text{O}_{13-\delta}$ as reported in reference [1]. This measured conductivity indicates that the prepared $\text{Sr}_4\text{Fe}_6\text{O}_{13-\delta}$ film in our work is a mixed oxygen and electron conductor.

It is worth noting that the direct measurement of these local tt-APBs is extremely difficult for most of the characterization techniques of ionic conductivity by using impedance spectroscopy, due to limited spatial resolution. Besides, The electric field for the in-situ experiment at electron microscopy is not easy to apply in this FIB lift-out sample due to the lack of a bottom electrode. Meanwhile, LaSrMnO_3 or SrRuO_3 et al., commonly used as oxide bottom electrodes are not easy to be integrated into this $\text{Sr}_4\text{Fe}_6\text{O}_{13-\delta}$ epitaxial films, since additional optimization of growth conditions of Pulse Laser Deposition is required. On the other hand, the ionic

conductivities of lithium ions and oxygen ions in perovskite structure calculated by classic MD simulations have been shown a good agreement with the ones observed in experiments (Ref: [1] Y. Yamamura et al. (2003). *Solid State Ionics* 160, 93-101; [2] Chen et al., (2015). *J. Am. Ceram. Soc.*, 98(2), 534–542), which indicates MD simulation can be served as an effective way to investigate the ionic conductivity result from ion diffusion. Therefore, MD simulations were performed in this study to understand the effect of tt-APB on ionic conductivity. To support our interpretation of oxygen migration, we provided additional analysis on oxygen migration at tt-APBs. As shown in Figure R4 extracted from Figure 4, the white circles highlight the variation of local iron distances due to the oxygen migration. It can be found that the shrinking features can be formed due to the oxygen deficiency at the time of 582 s, but then the shrinking features disappear due to the filling of extra oxygen atoms at the time of 670 s. Finally, the shrinking features appear again at the time of 838 s. Therefore, this dynamic oxygen migration can be indeed observed at tt-APBs. In comparison, we also employed an electron beam (with low beam current as in Figure R8) to irradiate the double Fe-O layers in $\text{Sr}_4\text{Fe}_6\text{O}_{13-\delta}$ as shown in Figure R5, it can be found that there is no obvious structure change during the continuous beam excitation. This direct experimental evidence further demonstrates that oxygen atoms can migrate across the tt-APBs.

Figure R3. The measurement of total conductivity in prepared $\text{Sr}_4\text{Fe}_6\text{O}_{13-\delta}$ film. (a) The plot of representative impedance spectra measured at the different temperatures for the prepared films. (b) The plot of the real part of the lateral ionic conductivity versus frequency for the prepared film. (c) Lateral ionic conductivity versus inverse temperature.

Figure R4. The evolution of local lattice structure at tt-APBs after being excited by the electron beam.

Figure R5. The evolution of lattice structure at Fe-O layer in $\text{Sr}_4\text{Fe}_6\text{O}_{13-\delta}$ after being excited by the electron beam. The experiment setup is the same as in Figure 4.

Revision to question 4:

The following information is added in the Introduction part, line 72.

‘The ionic conductivities of lithium ions and oxygen ions in perovskite structure calculated by classical MD simulations have been shown a good agreement with the ones observed in experiments, which indicates MD simulation can be served as an effective way to investigate the ion conductivity result from ion diffusion.’

The following information is added in the Materials and Methods part, line 3771.

‘In-plane AC impedance measurements were carried out by sputtering platinum interdigitated electrode (150 μm space, and 2000 μm arm length) onto the film surface. The AC impedance measurements were performed in the frequency range of 20 Hz - 10 MHz by an HP 4990A Precision Impedance Analyser with 50 mV voltage. Variable temperature measurements between 300 K and 673 K were realized by a probe station equipped with a hot plate.’

The following information in Figure R4 is added in Fig. S8.

Fig. S8. The evolution of lattice structure at Fe-O layer in $\text{Sr}_4\text{Fe}_6\text{O}_{13-\delta}$ after being excited by the electron beam. The experiment setup is the same as in Figure 4.

The following information in Figure R5 is added in Figure S1 and the caption of Figure S1 is revised.

(b) The plot of representative impedance spectra measured at the different temperatures for the prepared films. (c) The plot of the real part of the lateral ionic conductivity versus frequency for the prepared film. (d) The lateral ionic conductivity versus inverse temperature.

(5) Some minor points:

Please add the sample names in Figure 1a for left and right side.

In Figure 2c and 2d, the vertical axis needs to be labeled with exact numbers.

Response: Thanks for pointing this out. We have revised it in Figures 1 and 2.

Revision to question 5:

The following information highlighted in yellow color is added in Figure 1a.

The additional information is added and revised in Figure 2c, d.

Fig. 2. Atomic-scale composition and orbital occupation state analysis of topotactically transformable APB-III. **a**, Experimental and simulated high angle annular dark field scanning transmission electron microscopy (HAADF STEM) images of four Fe-O layers with a zigzag structure at topotactically transformable APB-III. **b**, HAADF STEM image and energy dispersive X-ray spectroscopy (EDXS) maps of Fe and Sr elements at APB-III. **c**, **d**, Spectra of Fe $L_{2,3}$ edges and normalized O K -edges acquired from the different Fe-O layers as shown in the corresponding HAADF STEM images on top. Distorted FeO_6 is referred to the octahedral Fe-O layer within perovskite layers of $\text{Sr}_4\text{Fe}_6\text{O}_{13-\delta}$. Undistorted FeO_6 is referred to the octahedral Fe-O layer within perovskite layers of APB-III. 1-FeO_4 of $\text{Sr}_4\text{Fe}_6\text{O}_{12}$ and 2-FeO_4 of APB-III represent the regions consisting of two and four tetrahedral layers, respectively, while $\text{FeO}_{4+\delta}$ represents the regions consisting of two pentahedral layers in $\text{Sr}_4\text{Fe}_6\text{O}_{13-\delta}$. Due to the slight variation of oxygen content, this layer may have FeO_4 or FeO_5 , so $\text{FeO}_{4+\delta}$ is used here. The feature of the shoulder marked by a black arrow can be resolved in the Fe L_3 edge of undistorted FeO_6 layers.

Reviewer #2 (Remarks to the Author):

The manuscript describes the atomic-scale observation of topotactically transformable antiphase boundaries of strontium ferrite oxide epitaxial thin films and the molecular dynamics simulations of the temperature dependence of ionic conductivity. The detailed observations and supporting simulation results are impressive. I agree with the claim that this study will open a new path of defect engineering strategies for improving ionic transportation by utilizing topotactic transformation. However, in the current version, there are some points that need to be considered, see below:

Response: We would like to express our gratitude to the reviewer for the positive recognition of our work and constructive comments. We also gratefully thank to referee for the valuable suggestions, which helped us improve the quality and clarity of the revised manuscript.

1. "antiphase boundaries (APBs)" is too technical a term to understand. It is advisable to include an explanation of "APBs" in the Introduction section.

Response: We appreciate for referee's insightful consideration and constructive suggestion on our incomplete introduction part. We fully agree with the referee's opinion about the additional description of antiphase boundaries. So, this information related to the explanation of antiphase boundaries is provided and highlighted in the revised introduction part.

Revision to question 1:

The following information is added in the introduction, line 51.

‘Among the various types of extended defects, the antiphase boundaries (APB) are commonly observed in ferrite films¹⁹. This boundary is defined as the interface between adjacent regions, where these regions exhibit a few sub-unit-cell shifts relative to each other along the specific crystallographic direction, e.g. out-of-plane direction, in the adjacent domains. These phase boundaries disrupt the continuity of the layered structure in the in-plane direction and exert a significant influence on the material properties.^{19, 20, 21}’

2. Fig. 1a contains the magnified views of the TEM images, but the caption does not. The caption should be correctly described.

Response: Thanks for pointing this out. We have revised it.

Revision to question 2:

The following information is revised in the caption of Figure 1a, line 106.

‘Fig. 1. Comparison of the atomic structure of topotactically transformable APBs and topotactically non-transformable APBs. a, The magnified view of high angle annular dark field scanning transmission electron microscopy- (HAADF STEM) images of $\text{Sr}_4\text{Fe}_6\text{O}_{12}$ and $\text{Sr}_4\text{Fe}_6\text{O}_{13}$, with different oxygen content, together with corresponding schematic images of atomic models of $\text{Sr}_4\text{Fe}_6\text{O}_{12}$ and $\text{Sr}_4\text{Fe}_6\text{O}_{13}$. By annealing in an oxidized or reduced atmosphere, double Fe layers with different topotactic phases can be transformed reversibly.’

3. One of the most important claims of this paper are that the zigzag four Fe-O layers at APB-III can be transformed before and after fully being oxidized, as shown in Fig. 1d and Fig. S4. However, the reduction and re-oxidation methods are not described in the Methods section. It is very important for this paper to specify these methods.

Response: Thanks for pointing this out. We acknowledge the reviewer's suggestion. So, this information related to the details of the reduction and re-oxidation methods is provided and highlighted in the revised Materials and Methods part.

Revision to question 3:

The following information is added in the Materials and Methods part, line 366.

'The reduction and re-oxidation processes were completed using TEM samples directly as shown in Fig. S6. The re-oxidation process was achieved by annealing the TEM sample in a high oxygen partial pressure environment ($P(O_2)$ of ~ 50 Pa) at 400 °C for 4 hours. The warming and cooling rates were both 10 °C/min. The reduction process was achieved by annealing the TEM samples at 400 °C for 4 hours. The TEM samples were placed in the crucible, which was filled with CaH_2 powder. The crucible was covered by aluminum foil.'

The following information is added in the Supplementary Materials, Figure S6.

'Fig. S6. The schematic illustration of the TEM sample annealed in a reduced/oxidized atmosphere. (a) Schematic model of the annealing process of reduction and re-oxidation for TEM samples. (b) A cross-sectional TEM sample is fixed at the Cu grid by using a focused ion beam, and then the TEM specimen can be annealed in an oxygen-enriched atmosphere. (c) Finally, the oxidized cross-sectional sample can be observed in the TEM instrument.'

4. Why is the in-plane ionic conductivity greatest at 550 K? Why does ionic conductivity decrease at higher temperatures? There should be more discussion about the relationship between temperature and ionic conductivity.

Response: Thanks for pointing this out. To better interpret the relationship between temperature and ionic conductivity, we modify Fig 5c,d,e and add the oxygen ion density at 400 K. The

localized displacements of oxygen ions in $\text{Sr}_4\text{Fe}_6\text{O}_{13}$ at both 400 and 550 K (Fig. 5d) confirm that the oxygen ions in $\text{Sr}_4\text{Fe}_6\text{O}_{13}$ are very stable and can only move locally. The broader distribution of oxygen ion density along the in-plane direction of the topotactically transformable APBs with four Fe-O layers at 550 K than that at 400 K (Fig 5e) indicates the increasing in-plane ionic conductivity with temperature is mainly attributed to the thermally activated process. For $\text{Sr}_4\text{Fe}_6\text{O}_{12}$, the structural difference at 400 K and 550K can be observed (Fig 5c), which is consistent with the neutron powder diffraction data that $\text{Sr}_4\text{Fe}_6\text{O}_{12}$ is unstable when the temperature is near 600 K. (Ref: Lü et al. (2013), *Angew Chem*, 125, 4933-4936) The structural change at 550 K makes the oxygen ions difficult to move in the in-plane direction, which can be seen from the oxygen density in Fig. 5c, leading to the degraded in-plane ionic conductivity comparing with that at 400 K.

Revision to question 4:

The following information is revised in Figure 5.

The paragraph in the results and discussion, line 314, is revised accordingly, and the following information is added in line 326.

‘The broader distribution of oxygen ion density along the in-plane direction of the topotactically transformable APBs with four Fe-O layers at 550 K than that at 400 K (Fig 5e) indicates the increasing in-plane ionic conductivity with temperature is mainly attributed to the thermally activated process. For $\text{Sr}_4\text{Fe}_6\text{O}_{12}$, the structural difference at 400 and 550K can be observed (Fig 5d), which is consistent with the neutron powder diffraction data that $\text{Sr}_4\text{Fe}_6\text{O}_{12}$ is unstable when temperature is near 600 K.⁴⁷ The structural change at 550 K makes the oxygen ions difficult to move in the in-plane direction, which can be seen from the oxygen density in Fig. 5d, leading to the degraded in-plane ionic conductivity comparing with that at 400 K.’

5. In Fig. S1, the caption should be changed because the coherency of the film cannot be determined from this XRD pattern alone. Note that the thin-film coherency is fully evident from the TEM images.

Response: Thanks for pointing this out. We acknowledge the reviewer's suggestion. We have revised these misleading interpretations throughout the manuscript.

Revision to question 5:

The following information is revised in the caption of Figure S1.

'Fig. S1. X-ray Diffraction characterization of as-grown $\text{Sr}_4\text{Fe}_6\text{O}_{13-8}$ films (a) X-ray θ - 2θ diffraction patterns of prepared epitaxial film on a LaAlO_3 substrate, where no additional peaks are observed, verifying that epitaxial film maintains coherency with the substrate.'

6. In Fig. S2, the difference between (a,b) and (c,d) is not clear from the caption. In addition, the caption should state the substance name.

Response: Thanks for pointing this out. We have revised these misleading interpretations in the caption of Figure S2.

Revision to question 6:

The following information is revised in the caption of Figure S2.

'Fig. S2. Low-magnified cross-sectional HAADF STEM images of prepared $\text{Sr}_4\text{Fe}_6\text{O}_{13-8}$ films. Experimental HAADF STEM atomic image of prepared $\text{Sr}_4\text{Fe}_6\text{O}_{13-8}$ films in the (a) region I and (c) the region II viewed along the cross-sectional direction. Regions with APBs-I are highlighted by white dotted arrows. It can be found that multiple APBs-I can be parallel or nearly symmetric to each other. Sr atom positions in the images of the (b) region I and the (d) region II are marked with yellow circles, to highlight the interfaces of APBs between adjacent domains with the atomic layer shifting against each other along out-of-plane direction.'

7. In Fig. S4, the caption should state the substance name.

Response: Thanks for pointing this out. We are sorry for our misleading and vague interpretation in this caption. The caption in Fig. S4 has been revised.

Revision to question 7:

The following information is revised in the caption of Figure S4.

‘Fig. S4. Atomic-scale composition and structure analysis of four Fe-O layers at the topotactically transformable APBs-III, IV in as-prepared $\text{Sr}_4\text{Fe}_6\text{O}_{13-\delta}$ films. (a) Atomic-scale EDXS element mapping of four Fe-O layers after being oxidized. (b) Experimental and simulated ABF, HAADF STEM images of four Fe-O layers after being oxidized in as-prepared $\text{Sr}_4\text{Fe}_6\text{O}_{13-\delta}$ films. (c) HADDF STEM images of four Fe-O layers at the topotactically transformable APBs-III, IV before and after being oxidized in as-prepared $\text{Sr}_4\text{Fe}_6\text{O}_{13-\delta}$ films. Color maps on the left side refer to the distribution of lateral inter-atomic spacing (defined in the left schematic inset).’

8. Minor points and typos:

Line 85: "High" change to "high"

Lines 244, 255, 257: "topotacitcally" change to "topotactically"

Line 306: Subscript of chemical formula

Response: Thanks for pointing this out. We really appreciate for referee's careful inspection, which helped us improve the quality and clarity of the revised manuscript. We have corrected these misleading interpretations.

Revision to question 8:

The following information is revised in the result and discussion part, line 95.

‘The ~~H~~high angle annular dark field (HAADF) STEM image in the left side of Fig. 1a shows the atomic structure of as-grown films, which is close to the crystalline structure of $\text{Sr}_4\text{Fe}_6\text{O}_{13}$ with higher oxygen content.’

The following information is revised in the result and discussion part, line 266, 270, 282.

‘Topotacitcally’ has been revised as ‘Topotactically’

The following information is revised in the result and discussion part, line 338.

‘Fig. 5. Temperature dependence of ionic conductivity of $\text{Sr}_4\text{Fe}_6\text{O}_{12}$, $\text{Sr}_4\text{Fe}_6\text{O}_{13}$, and the topotactically transformable APBs-III with four Fe-O layers structures.’

Reviewer #3 (Remarks to the Author):

The paper describes details of the topotactic defects in Sr-Fe-O oxides and discusses their effects on the ionic conductivity. Various type of the local defects, termed as being of antiphase-boundary type (APB), have been successfully identified based on atomic-resolution microscopy analysis, and the occurrence of such variations are reasonably related with the local oxygen contents, as investigated by microbeam spectroscopy. Overall the manuscript is well organized, but I find that quantitative estimations are lacking in local oxygen imaging as well as spectroscopy analysis. In addition, there could be a problematic point on the supposed oxygen behavior during in-situ beam-irradiating observations. Because of these, I cannot recommend the present manuscript for publication in Nature Communications. Detailed comments are described below.

Response: We would like to express our gratitude to the reviewer for the positive recognition of our work and constructive comments. We also gratefully thank to referee for the valuable suggestions, which helped us improve the quality and clarity of the revised manuscript. We have revised some interpretations and provided additional quantitative analysis of local oxygen imaging to support our conclusion.

1. Quantitative analysis is needed for both oxygen atom imaging and local oxygen content. A series of dark-field images of cation sites are convincing and well supported by simulations. On the other hand, bright-field imaging of oxygen atoms is not clear enough and hence not convincing, in particular for those located at interstitial sites (e.g, Fig. 3d). These should be shown in a more quantitative manner, e.g., intensity profile comparisons with the simulations. Also, the explanations on local oxygen contents by EELS profiles (Fig. 2c, d) are too qualitative. They should be evaluated quantitatively, by showing comparisons with the simulations and/or Fe/O ratio derived from each EELS profile.

Response: We gratefully thank to referee for the valuable comments and constructive suggestions. To give a quantitative analysis of bright field imaging of interstitial oxygen atoms, we have provided additional intensity profile comparison with the simulations as the referee suggested, and the additional data has been added in the updated Fig. 3b. As shown in the updated Figure 3b, the intensities of the line profile extracted from the atomic layers marked by the blue rectangle at APBs-III with and without interstitial oxygen atoms are plotted respectively. When compared with the simulated ABF image of APBs-III without interstitial oxygen atoms, the intensity of the line profile at the experimental ABF image at APBs-III with interstitial oxygen atoms shows two obvious small peaks (indicated by the blue arrows) at the interstitial sites between two large peaks corresponding to the iron positions. There are no obvious peaks in experimental and simulated ABF images at the APBs-III without interstitial oxygen atoms. The results are well consistent to support the conclusion that the interstitial oxygen atoms can be observed at APBs-III.

Besides, we also provide a more quantitative analysis of EELS in Fig. 2c, d. As shown in the newly added Figure S5, the ratios of integrated intensities of peak A and peak B are listed in the different FeO layers. It can be found that the I_A/I_B in 2-FeO₄ of tt-APBs is larger than other layers, which is consistent with the interpretation in the manuscript. the pre-peak in 2-FeO₄

exhibits much higher intensity than that in FeO_6 and $\text{FeO}_{4+\delta}$, and even higher than 1- FeO_4 layers due to the more interstitial sites than 1- FeO_4 layers. Double Fe-O layers (1- FeO_4 and $\text{FeO}_{4+\delta}$) also exhibit higher intensity in pre-peak compared with FeO_6 layers. To support our interpretation, the additional analysis on the oxygen K -edge is provided in Supplementary Materials Fig. S5. Besides, Fe/O ratios are shown in Figure R6 and Fig. S6.

Figure R6. The Fe/O ratio is plotted across the different layers of tt-APBs.

Revision to question 1:

The following information is added in Figure 3d, and the caption is revised.

‘Fig. 3. Atomic resolution imaging of interstitial oxygen atoms at the topotactically transformable APB. a, Schematic atomic model depicts the interstitial sites for oxygen atoms in the four Fe-O layers with a zigzag

structure. **b**, Experimental HAADF STEM and Annular bright-field scanning transmission electron microscopy (ABF STEM) images of four Fe-O layers with a zigzag structure before and after being partially oxidized. On the left side, the intensities of the line profile extracted from the regions marked by the blue rectangle are plotted. The blue arrows indicate the peaks of interstitial oxygen atoms. Some oxygen atoms can be found in interstitial sites of the zigzag region indicated by white arrows in ABF STEM image. **c**, Simulated HAADF and ABF STEM image of APBs-III without interstitial oxygen atoms. On the left side, the intensity of the line profile extracted from the regions marked by the blue rectangle is plotted. **d**, ABF STEM image of the zigzag APBs-III adjacent to FeO_{4+δ} double layers, clearly revealing interstitial oxygen atoms with significant contrast. The schematic picture is depicted in the right side. **e**, The discontinuous in-plane oxygen ionic channels blocked by the APB-I and APB-II marked by white arrows. **f**, APB-III shows continued oxygen ionic channel across the APBs, where four Fe-O layers at APB-III can be a bridge to connect the adjacent domains at the APBs.'

The following information is added in Fig. S5 of Supplementary Materials.

Fig. S5. The quantitative analysis of the ratios of integrated intensities of peak A and peak B.

The following information is added in Fig. S6 of Supplementary Materials.

‘Fig. S6. Atomic-scale composition analysis of topotactically transformable APB-III. (a) Experimental and simulated ABF STEM images along the [001] zone axis. (b) HAADF STEM image of zigzag structure with four Fe-O layers along [100] direction. The scale bar is 1 nm. (c) Fe element maps are extracted by using electron energy loss spectroscopy (EELS), and Fe/O relative content can be calculated as shown in the plot as the function of different layers of APB-III. The scale bar is 1 nm.’

2. Oxygen atom behaviors during the in-situ electron-beam radiations. The present in-situ beam-radiation experiments are aimed at promoting local oxygen diffusion. However, as is well known, irradiation with high-energy electron beams (acceleration voltage ~ 100 kV) is most likely to cause so-called ‘knock-on’ effects, which extract oxygen atoms from the specimen. This seems to be consistent with the results in Fig. 4b; change from $\text{FeO}_{4+\delta}$ to FeO_4 (i.e., oxygen deficient reaction). To prove that the observed structural change is indeed caused by the in-plane oxygen diffusion, the authors should show the oxygen increase at the neighboring regions, and confirm there is no change of total oxygen contents during the structure change including the neighbor regions (i.e., local decrease/increase oxygen atoms are indeed in a trade-off relation across the certain range).

Response: We gratefully thank to referee for the valuable comments and constructive suggestions. We agree with the reviewer that irradiation with high-energy electron beams (300 kV) is most likely to cause ‘knock-on’ effects, which can induce oxygen vacancies. Therefore, to minimize the damaging effect on our sample, we use extremely low beam currents (~ 10 pA) to do this in-situ observation of oxygen migration across the tt-APBs. The oxygen migration across the tt-APBs is a dynamic process where oxygen atoms might migrate from the tt-APBs

to inject into the adjacent regions where unfilled oxygen sites will accept these transferred oxygen atoms uniformly, however, EELS or HAADF image analysis is hard to detect this subtle oxygen variation in adjacent regions. To support our interpretation of oxygen migration, we provided additional analysis on oxygen migration at tt-APBs. As shown in Figure R7 extracted from Figure 4, the white circles highlight the variation of local iron distances due to the oxygen migration. It can be found that the shrinking features can be formed due to the oxygen deficiency at the time of 582 s, but then the shrinking features disappear due to the filling of extra oxygen atoms at the time of 670 s. Finally, the shrinking features appear again at the time of 838 s. Therefore, this dynamic oxygen migration can be indeed observed at tt-APBs. In addition, we also employed an electron beam (with low beam current as in Figure R8) to irradiate the double Fe-O layers in $\text{Sr}_4\text{Fe}_6\text{O}_{13-\delta}$ as shown in Figure R8, it can be found that there is no obvious structure change during the continuous beam excitation. So, the beam irradiation at low beam current cannot deplete the all of oxygen atoms at tt-APBs in Figure 4 due to the ‘knock-on’ effect. To better describe the dynamic process of oxygen migration at tt-APBs, we agree with the reviewer that irradiation includes the ‘knock-on’ effect as the excitation of oxygen migration. The initial fewer oxygen defects due to the ‘knock on’ effect of beam exposure with low beam current can be considered as the first empty sites. These interstitial oxygen atoms can migrate through these empty sites across the topotactically transformable APBs due to the energy perturbation of electron beam irradiation when the focused electron beam continuously irradiates the straight four Fe-O layers. Therefore, the ‘knock-on’ effect of electron beam irradiation can also be considered as the initial excitation for oxygen migration.

Figure R7. The evolution of lattice structure at tt-APBs after being excited by electron beam.

Figure R8. The evolution of lattice structure at Fe-O layer in $\text{Sr}_4\text{Fe}_6\text{O}_{13-\delta}$ after being excited by the electron beam. The experiment setup is the same as in Figure 4.

Revision to question 2:

The following information is added in the Methods and Materials part, line 389.

‘The beam current of 10 pA at STEM mode was employed to achieve the dynamic observation at tt-APBs.’

The following information in Figure R9 is added in Fig. S8.

Fig. S8. The evolution of lattice structure at Fe-O layer in $\text{Sr}_4\text{Fe}_6\text{O}_{13-\delta}$ after being excited by the electron beam. The experiment setup is the same as in Figure 4.

The following interpretation is revised in the Results and Discussions part, line 274.

‘The initial fewer oxygen defect due to the ‘knock on’ effect of beam exposure with low beam current can be considered as the first empty sites. And these interstitial oxygen atoms can migrate through these empty sites across the topotactically transformable APBs due to the energy perturbation of electron beam irradiation when the focused electron beam continuously irradiates the straight four Fe-O layers.^{26,45} In contrast, ‘knock on’ effect induced by electron beam irradiation cannot easily lead to the transformation from $\text{Sr}_4\text{Fe}_6\text{O}_{13}$ to $\text{Sr}_4\text{Fe}_6\text{O}_{12}$ at the same experimental condition (Seen in Supplementary Fig. 8).’

Reviewer #4 (Remarks to the Author):

This manuscript presents the first atomic-resolution visualization of topotactical transformation occurred at antiphase boundary (APB) in epitaxial $\text{Sr}_4\text{Fe}_6\text{O}_{12+\delta}$ films for using as mixed ionic-electronic conductors (MIECs). Using a high-resolution scanning transmission electron microscopy (STEM) combined with electron energy loss spectroscopy (EELS), the authors showed the high-quality high-angle annular dark field (HAADF) STEM and annular bright field (ABF) STEM images of various antiphase boundaries with clearly identifying Fe and oxygen individual atoms. Because probing the oxygen migration channels for optimizing the ionic conductivity has been very important and challenging issue in the solid oxide fuel cell field, this work is so unique and interesting with some very excellent STEM imaging presented. In addition, the authors suggested special type of APBs with the capability of topotactic transformation showing dynamic changes of oxygen ions while annealing in an oxidation condition and irradiating an electron beam during STEM observation. This work is very interesting finding and the results are so clear. The manuscript is well written and organized. Therefore, I recommend this manuscript to be published in this journal after addressing the minor comments.

Response: We thank the Referee for the highly positive opinion about the importance of the topic addressed in our manuscript.

1. The authors showed the migration and diffusion of oxygen ions during oxidation (Fig. 3) and e-beam irradiation (Fig. 4). Because the STEM results can be dependent on the annealing and e-beam irradiation conditions, the detailed experimental procedures should be described in METHOD part for the reader.

Response: Thanks for pointing this out. We acknowledge the reviewer's suggestion. So, this information related to the details of the reduction and re-oxidation methods is provided and highlighted in the revised Materials and Methods part and Supplementary Materials.

Revision to question 1:

The following information is added in the Materials and Methods part, line 366.

'The reduction and re-oxidation processes were completed using TEM samples directly. The re-oxidation process was achieved by annealing the TEM sample in a high oxygen partial pressure environment ($P(\text{O}_2)$ of ~ 50 Pa) at 400°C for 4 hours. The warming and cooling rates were both $10^\circ\text{C}/\text{min}$. The reduction process was achieved by annealing the TEM sample at 400°C for 4 hours. The TEM sample was placed in the crucible, which was filled with CaH_2 powder. The crucible was covered by aluminum foil. In-plane AC impedance measurements were carried out by sputtering platinum interdigitated electrode ($150\ \mu\text{m}$ space, and $2000\ \mu\text{m}$ arm length) onto the film surface. The AC impedance measurements were performed in the frequency range of $20\ \text{Hz} - 10\ \text{MHz}$ by an HP 4990A Precision Impedance Analyser with $50\ \text{mV}$ voltage. Variable temperature measurements between $300\ \text{K}$ and $673\ \text{K}$ were realized by a probe station equipped with a hot plate.'

The following information is added in the Supplementary Materials, Figure S6.

‘Fig. S6. The schematic illustration of the TEM sample annealed in a reduced/oxidized atmosphere. (a) Schematic model of the annealing process of reduction and re-oxidation for TEM samples. (b) A cross-sectional TEM sample is fixed at the Cu grid by using a focused ion beam, and then the TEM specimen can be annealed in an oxygen-enriched atmosphere. (c) Finally, the oxidized cross-sectional sample can be observed in the TEM instrument.’

The following information is added in the Methods and Materials part, line 389.

‘The beam current of 10 pA at STEM mode was employed to achieve the dynamic observation at tt-APBs.’

The following information in Figure R3 is added in Fig. S8.

Fig. S8. The evolution of lattice structure at Fe-O layer in $\text{Sr}_4\text{Fe}_6\text{O}_{13-\delta}$ after being excited by the electron beam. The experiment setup is the same as in Figure 4.

The following interpretation is revised in the Results and Discussions part, line 274.

‘The initial fewer oxygen defect due to the ‘knock on’ effect of beam exposure with low beam current can be considered as the first empty sites. And these interstitial oxygen atoms can migrate through these empty sites across the topotactically transformable APBs due to the energy perturbation of electron beam irradiation’

when the focused electron beam continuously irradiates the straight four Fe-O layers.^{26,45} In contrast, 'knock on' effect induced by electron beam irradiation cannot easily lead to the transformation from $\text{Sr}_4\text{Fe}_6\text{O}_{13}$ to $\text{Sr}_4\text{Fe}_6\text{O}_{12}$ at the same experimental condition (Seen in Supplementary Fig. 8). '

Reviewers' Comments:

Reviewer #1:

Remarks to the Author:

The authors have answered my questions.

Although the direct experimental evidence for enhanced ionic conductivity is not included, and the discussion is based on MD simulation, the findings are important enough. Many researchers would find it interesting and I think the manuscript is worthy to be published on Nature Communications.

Reviewer #2:

Remarks to the Author:

The authors have properly answered my questions and comments. I thus recommend the publication in Nature Communications.

Reviewer #3:

Remarks to the Author:

The authors have addressed several concerns I have noted and revised the manuscript accordingly. I recommend publication of this manuscript.

Reviewer #4:

Remarks to the Author:

The authors have properly revised their manuscript based on reviewer's comments and suggestions.

As far as I am concerned, the manuscript can be published in Nature communications.

Responses to comments

We gratefully thank the editor and all reviewers for the valuable comments and constructive suggestions, which substantially helped us improve the quality and clarity of the revised manuscript. The point-to-point responses to all the comments of the reviewers are listed in detail as follows.

REVIEWER COMMENTS

Reviewer #1 (Remarks to the Author):

The authors have answered my questions.

Although the direct experimental evidence for enhanced ionic conductivity is not included, and the discussion is based on MD simulation, the findings are important enough. Many researchers would find it interesting and I think the manuscript is worthy to be published on Nature Communications.

Response: Thanks for the reviewer's positive comments on this manuscript. We gratefully thank to referee for the valuable comments and constructive suggestions, which helped us improve the quality and clarity of the revised manuscript.

Reviewer #2 (Remarks to the Author):

The authors have properly answered my questions and comments. I thus recommend the publication in Nature Communications.

Response: We would like to express our gratitude to the reviewer for the positive recognition of our work and constructive comments. We also gratefully thank to referee for the valuable suggestions, which helped us improve the quality and clarity of the revised manuscript.

Reviewer #3 (Remarks to the Author):

The authors have addressed several concerns I have noted and revised the manuscript accordingly. I recommend publication of this manuscript.

Response: We would like to express our gratitude to the reviewer for the positive recognition of our work and constructive comments. We also gratefully thank to referee for the valuable suggestions, which helped us improve the quality and clarity of the revised manuscript.

Reviewer #4 (Remarks to the Author):

The authors have properly revised their manuscript based on reviewer's comments and suggestions. As far as I am concerned, the manuscript can be published in Nature

communications.

Response: We thank the Referee for the highly positive opinion about the importance of the topic addressed in our manuscript.